# Differential adhesion regulates neurite placement via a retrograde zippering mechanism

Titas Sengupta[1], Noelle L Koonce[1], Nabor Vázquez-Martínez[1], Mark W Moyle[1], Leighton H Duncan[1], Sarah E Emerson[1], Xiaofei Han[2], Lin Shao[1], Yicong Wu[2], Anthony Santella[3], Li Fan[3], Zhirong Bao[3], William A Mohler[4], Hari Shroff[2,5], Daniel A Colón-Ramos[1,5,6,7]*

[1]Department of Neuroscience and Department of Cell Biology, Yale University School of Medicine, New Haven, United States; [2]Laboratory of High Resolution Optical Imaging, National Institute of Biomedical Imaging and Bioengineering, National Institutes of Health, Bethesda, United States; [3]Developmental Biology Program, Sloan Kettering Institute, New Haven, United States; [4]Department of Genetics and Genome Sciences and Center for Cell Analysis and Modeling, University of Connecticut Health Center, Farmington, United States; [5]MBL Fellows, Marine Biological Laboratory, Woods Hole, United States; [6]Wu Tsai Institute, Yale University, New Haven, United States; [7]Instituto de Neurobiología, Recinto de Ciencias Médicas, Universidad de Puerto Rico, San Juan, Puerto Rico

*For correspondence:
daniel.colon-ramos@yale.edu

Competing interest: The authors declare that no competing interests exist.

**Abstract** During development, neurites and synapses segregate into specific neighborhoods or layers within nerve bundles. The developmental programs guiding placement of neurites in specific layers, and hence their incorporation into specific circuits, are not well understood. We implement novel imaging methods and quantitative models to document the embryonic development of the *C. elegans* brain neuropil, and discover that differential adhesion mechanisms control precise placement of single neurites onto specific layers. Differential adhesion is orchestrated via developmentally regulated expression of the IgCAM SYG-1, and its partner ligand SYG-2. Changes in SYG-1 expression across neuropil layers result in changes in adhesive forces, which sort SYG-2-expressing neurons. Sorting to layers occurs, not via outgrowth from the neurite tip, but via an alternate mechanism of retrograde zippering, involving interactions between neurite shafts. Our study indicates that biophysical principles from differential adhesion govern neurite placement and synaptic specificity in vivo in developing neuropil bundles.

## Editor's evaluation

Your work provides novel and interesting insights into circuit formation, demonstrating how synaptic specificity is controlled at least in part by different cell adhesion during neurite placement. The revisions of your paper have addressed the points raised by the reviewers and we are glad to see that those revisions have further strengthened the conclusion of this paper.

## Introduction

In brains, neuronal processes or neurites are segregated away from cell bodies into synapse-rich regions termed neuropils: dense structures of nerve cell extensions which commingle to form

functional circuits (*Maynard, 1962*). In both vertebrates and invertebrates, placement of neurites into specific neighborhoods results in a laminar organization of the neuropil (*Kolodkin and Hiesinger, 2017*; *Millard and Pecot, 2018*; *Nevin et al., 2008*; *Sanes and Zipursky, 2010*; *Schurmann, 2016*; *Soiza-Reilly and Commons, 2014*; *Xu, 2020*; *Zheng et al., 2018*). The laminar organization segregates specific information streams within co-located circuits and is a major determinant of synaptic specificity and circuit connectivity (*Baier, 2013*; *Gabriel et al., 2012*; *Missaire and Hindges, 2015*; *Moyle et al., 2021*; *Nguyen-Ba-Charvet and Chédotal, 2014*; *White et al., 1986*; *Xie et al., 2017*). The developmental programs guiding placement of neurites along specific layers, and therefore circuit architecture within neuropils, are not well understood.

The precise placement of neurites within layered structures cannot be exclusively explained by canonical tip-directed outgrowth dynamics seen during developmental axon guidance (*Tessier-Lavigne and Goodman, 1996*). Instead, ordered placement of neurites resulting in layered patterns appears to occur via local cell-cell recognition events. These local cell-cell recognition events are modulated by the regulated expression of specific cell adhesion molecules (CAMs) that place neurites, and synapses, within nerve bundles (*Aurelio et al., 2003*; *Kim and Emmons, 2017*; *Lin et al., 1994*; *Petrovic and Hummel, 2008*; *Poskanzer et al., 2003*; *Schwabe et al., 2019*). For example, studies in both the mouse and fly visual systems have revealed important roles for the regulated spatio-temporal expression of IgSF proteins, such as Sidekick, Dscam and Contactin, in targeting synaptic partner neurons to distinct layers or sublayers (*Duan et al., 2014*; *Sanes and Zipursky, 2010*; *Tan et al., 2015*; *Yamagata and Sanes, 2008*; *Yamagata and Sanes, 2012*). In *C. elegans* nerve bundles, neurite position is established and maintained via combinatorial, cell-specific expression of CAMs which mediate local neurite interactions and, when altered, lead to defects in neurite order within bundles (*Kim and Emmons, 2017*; *Yip and Heiman, 2018*). How these local, CAM-mediated interactions are regulated during development and how they result in the segregation of neurites into distinct layers, are not well understood.

Differential expression of cell adhesion molecules (CAMs) in undifferentiated cells from early embryos can drive their compartmentalization (*Foty and Steinberg, 2005*; *Foty and Steinberg, 2013*; *Steinberg, 1962*; *Steinberg, 1963*; *Steinberg, 1970*; *Steinberg and Takeichi, 1994*). This compartmentalization is in part regulated by biophysical principles of cell adhesion and surface tension which can give rise to tissue-level patterns and boundaries (*Canty et al., 2017*; *Duguay et al., 2003*; *Erzberger et al., 2020*; *Foty et al., 1996*; *Schötz et al., 2008*). Morphogenic developmental processes such as the patterning of the *Drosophila* germline and retina, the germ layer organization in zebrafish, and the sorting of motor neuron cell bodies into discrete nuclei in the ventral spinal cord can be largely explained via differential adhesion mechanisms and cortical contraction forces that contribute to cell sorting (*Bao and Cagan, 2005*; *Bao et al., 2010*; *Godt and Tepass, 1998*; *González-Reyes and St Johnston, 1998*; *Krieg et al., 2008*; *Price et al., 2002*; *Schötz et al., 2008*). While differential adhesion is best understood in the context of the sorting of cell bodies in early embryogenesis, recent neurodevelopmental work supports that this mechanism influences sorting of neuronal processes in vivo as well. For example, differential expression of N-cadherin in the *Drosophila* visual system underlies the organization of synaptic-partnered neurites (*Schwabe et al., 2019*), where changes in the relative levels of N-cadherin are sufficient to determine placement of neurites within nerve bundles. Whether differential adhesion acts as an organizational principle within layered neuropils and how it regulates precise placement of neurites is not known.

Here, we examine the developmental events that lead to placement of the AIB interneurons in the *C. elegans* nerve ring. The *C. elegans* nerve ring is a layered neuropil, with specific layers or strata functionally segregating sensory information and motor outputs (*Brittin et al., 2021*; *Moyle et al., 2021*; *White et al., 1986*). A highly interconnected group of neurons referred to as the 'rich club' neurons, and which include interneuron AIB, functionally link distinct strata via precise placement of their neurites (*Moyle et al., 2021*; *Sabrin, 2019*; *Towlson et al., 2013*). Each AIB interneuron projects a single neurite, but segments of that single neurite are placed along distinct and specific layers in the *C. elegans* nerve ring (*Figure 1*). The sequence of events resulting in the precise placement of AIB along defined nerve ring layers is unexplored, primarily owing to limitations in visualizing these events in vivo during embryonic stages.

We implemented novel imaging methods and deep-learning approaches to yield high-resolution images of AIB during embryonic development. We discovered that placement of the AIB neurite

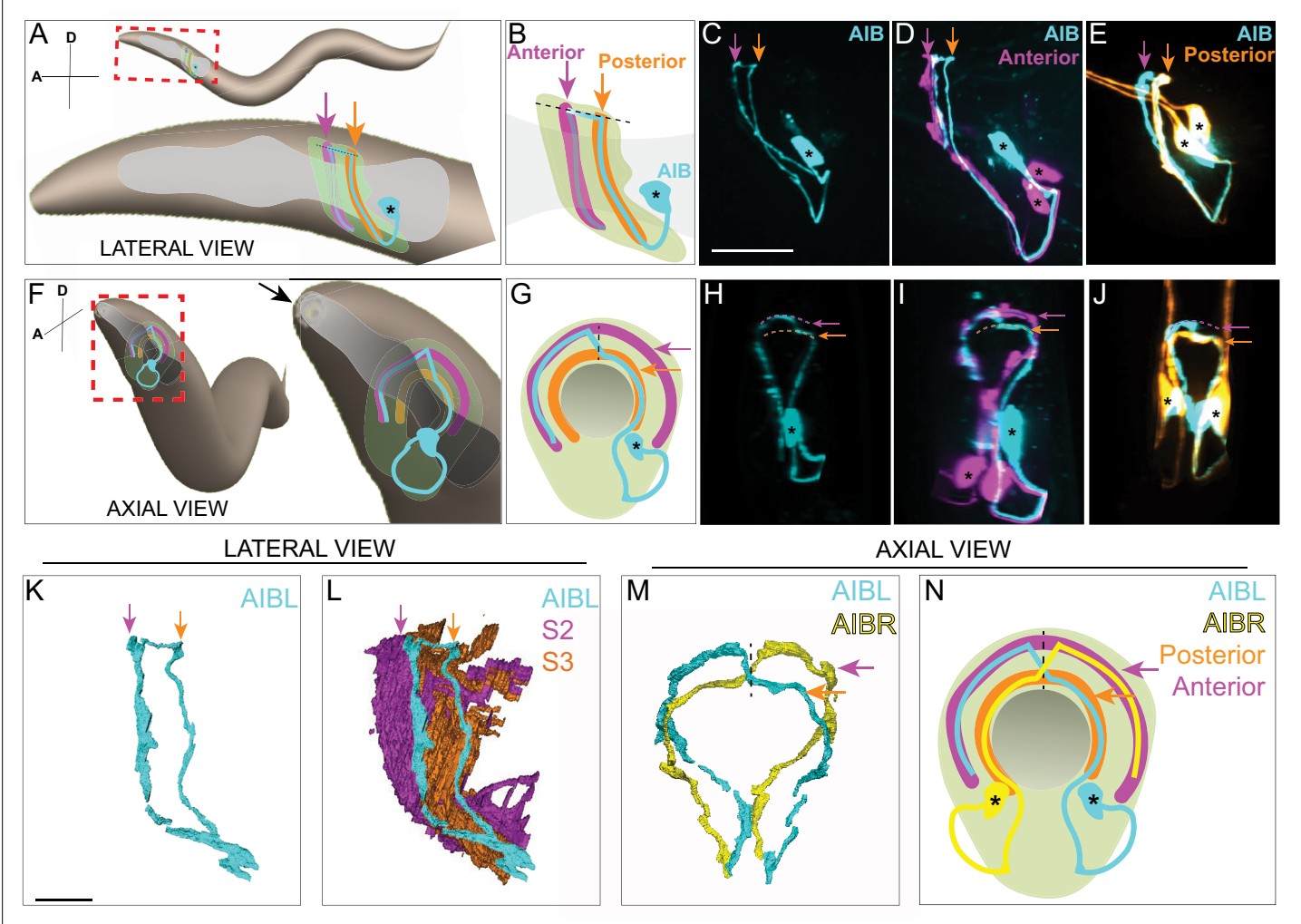

**Figure 1.** AIB single neurite is placed along two distinct neighborhoods in the nerve ring. (**A**) Schematic of an adult/larval *C. elegans* showing an AIB neuron (cyan) and its posterior (orange) and anterior (magenta) neighborhoods in the head. The AIB neurite has a proximal neurite segment (orange arrow), a posterior-anterior shift at the dorsal midline (dashed line) and a distal neurite segment (magenta arrow; on the other side of the worm, behind the pharynx, which is in gray). The neon-colored outline represents the nerve ring neuropil. The terms 'proximal' or 'distal' neurite segments refer to the relationship of the neurite segment to the AIB cell body. The neighborhoods in which the 'proximal' and 'distal' neurite segments are positioned are referred to as the 'posterior' or 'anterior' neighborhoods, respectively, because of their position along the anterior-posterior axis of the worm. Note that this schematic only shows one neuron of the AIB pair. Cell body is marked with an asterisk. (**B**) Magnified schematic of AIB and its neighborhoods in (**A, C**) Representative confocal image showing the lateral view of an AIB neuron labeled with cytoplasmic mCherry (cyan). (**D**) Representative confocal image showing an AIB neuron labeled with cytoplasmic mCherry (cyan); and RIM motor neuron of the anterior neighborhood labeled with cytoplasmic GFP (magenta) in lateral view. Note the colocalization of the AIB distal neurite (but not the proximal neurite) with the anterior neighborhood marker RIM (compare with **E**). (**E**) As (**D**), but with AIB (cyan) and AWC and ASE sensory neurons of the posterior neighborhood (orange). Note the colocalization of the AIB proximal neurite (but not the distal neurite) with the posterior neighborhood markers AWC and ASE (compare with **D**). (**F–J**) Same as **A–E** but in axial view indicated by the arrow in (**F**). The worm head is tilted in this view to make the two neurite segments in the two neighborhoods visible. Note shift in **H** (arrows), corresponding to AIB neurite shifting neighborhoods (compare **I** and **J**). (**K,L**) Volumetric reconstruction from the JSH electron microscopy connectome dataset (*White et al., 1986*) of AIBL (**K**), and AIBL overlaid on nerve ring strata (**L**), in lateral view, with S2 and S3 strata (named as in *Moyle et al., 2021*), containing anterior and posterior neighborhoods, respectively. (**M**) Volumetric reconstruction of AIBL and AIBR in axial view (from the JSH dataset *White et al., 1986*). Note the shift in neighborhoods by AIBL and AIBR, at the dorsal midline (dashed line), forms a chiasm (also see *Figure 1—figure supplement 1*). (**N**) Schematic of **M** highlighting the AIB neighborhoods for context and the dorsal midline with a dashed line (AIB neighborhoods, synaptic polarity and resulting network properties also shown in *Figure 1—figure supplement 2*). Scale bar = 10 μm for **A–J** and 3 μm for **K–N**.

The online version of this article includes the following video, source data, and figure supplement(s) for figure 1:

**Figure supplement 1.** Neurite positions of the bilaterally symmetric AIB neurons.

**Figure supplement 2.** AIB contacts, synaptic distribution and network properties enable its function as a connector hub neuron.

*Figure 1 continued on next page*

*Figure 1 continued*

**Figure supplement 2—source data 1.** Pairwise cosine similarity values between the EM connectome datasets (*White et al., 1986*; *Witvliet et al., 2021*) analyzed in *Figure 1—figure supplement 2I*.

**Figure supplement 2—source data 2.** Betweenness centrality values for all neurons in the EM connectome datasets (*Witvliet et al., 2021*) analyzed in *Figure 1—figure supplement 2J*.

**Figure 1—video 1.** Neighborhoods of AIB, related to Figure 1, Figure 1—figure supplement 1 and Figure 1—figure supplement 2.
https://elifesciences.org/articles/71171/figures#fig1video1

**Figure 1—video 2.** AIB and anterior neighborhood neuron, RIM, related to Figure 1 and Figure 1—figure supplement 2.
https://elifesciences.org/articles/71171/figures#fig1video2

**Figure 1—video 3.** AIB and posterior neighborhood neurons, related to Figure 1 and Figure 1—figure supplement 2.
https://elifesciences.org/articles/71171/figures#fig1video3

**Figure 1—video 4.** The bilaterally symmetric AIBL and AIBR neurons, related to Figure 1 and Figure 1—figure supplement 1.
https://elifesciences.org/articles/71171/figures#fig1video4

**Figure 1—video 5.** Polarized localization of AIB presynaptic sites, related to Figure 1—figure supplement 2.
https://elifesciences.org/articles/71171/figures#fig1video5

depends on coordinated retrograde zippering mechanisms that align segments of the AIB neurite onto specific neuropil layers and is distinct from canonical tip-directed mechanisms of neurite placement. Quantitative analysis and modeling of our in vivo imaging data revealed that biophysical principles of differential adhesion influence the observed retrograde zippering mechanism that results in the sorting of the AIB neurite shaft onto distinct neuropil strata. We performed genetic screens to identify the molecular mechanisms underpinning these differential adhesion mechanisms, discovering a role for the IgCAM receptor *syg-1* and its ligand, *syg-2*. We determined that *syg-2* acts in AIB to instruct neurite placement across strata, while *syg-1* is required non-cell autonomously, and at specific layers. Temporally regulated expression of SYG-1 alters adhesive forces during development to sort segments of AIB onto specific layers. Ectopic expression of SYG-1 predictably affects differential adhesion across layers, repositioning the AIB neurite segments in a SYG-2-dependent manner. Our findings indicate that conserved principles of differential adhesion drive placement of neurites, and *en passant* synaptic specificity, in layered neuropils.

## Results

### Examination of AIB neurite architecture in the context of the nerve ring strata

First, we characterized the precise placement and synaptic distribution of the AIB neurite within the nerve ring neuropil strata. From electron microscopy connectome datasets and in vivo imaging, we observed that the AIB neurite is unipolar, with its single neurite placed along two distinct and specific strata of the nerve ring (*Figure 1*, *Figure 1—videos 1–3*).

Connectomic studies have identified AIB as a 'rich club' neuron, a connector hub that links nodes in different functional modules of the brain (*Sabrin, 2019*; *Towlson et al., 2013*). We observed that AIB's role as a connector hub was reflected in its architecture within the context of the layered nerve ring (*Figure 1K–N*, *Figure 1—figure supplement 1*, *Figure 1—figure supplement 2*). For example, the AIB neurite segment in the posterior neighborhood is enriched in postsynaptic specializations, enabling it to receive sensory information from the adjacent sensory neurons that reside in that neighborhood (*Figure 1—figure supplement 2*; *White et al., 1983*; *White et al., 1986*). AIB relays this sensory information onto the anterior neighborhood, where the AIB neurite elaborates presynaptic specializations that innervate neighboring motor interneurons (*Figure 1D, E1 and J*; *Figure 1—figure supplement 2A-G*, *Figure 1—video 5*). The architecture of AIB is reminiscent of that of amacrine cells of the inner plexiform layer (*Demb and Singer, 2012*; *Kolb, 1995*; *Kunzevitzky et al., 2013*; *Robles et al., 2013*; *Strettoi et al., 1992*; *Taylor and Smith, 2012*), which serve as hubs by distributing their neurites and synapses across distinct and specific sublaminae of the vertebrate retina (*Marc et al., 2014*). We set out to examine how this architecture was laid out during development.

# A retrograde zippering mechanism positions the AIB neurites in the anterior neighborhood during embryonic development

Prior to this study, using characterized cell-specific promoters, AIB could be visualized in larvae (*Altun and Chen, 2008*; *Kuramochi and Doi, 2018*) but not in embryos, when placement of AIB into the neighborhoods is specified (*Figure 1—figure supplement 2A* shows that by earliest postembryonic stage, L1, AIB neurite placement is complete, indicating placement occurs in the embryo). Moreover, continuous imaging of neurodevelopmental events in embryos, necessary for documenting AIB development, presents unique challenges regarding phototoxicity, speeds of image acquisition as it relates to embryonic movement, and the spatial resolution necessary to discern multiple closely spaced neurites in the embryonic nerve ring (*Wu et al., 2011*). These barriers prevented documentation of AIB neurodevelopmental dynamics. To address these challenges, we first adapted a subtractive labeling strategy for sparse labeling and tracking of the AIB neurites in embryos (detailed in Materials and methods, *Figure 2—figure supplement 2*,*Figure 2—video 1*, *Armenti et al., 2014*). We then adapted use of novel imaging methods, including dual-view light-sheet microscopy (diSPIM) (*Kumar et al., 2014*; *Wu et al., 2013*) for long-term isotropic imaging, and a triple-view line-scanning confocal imaging and deep-learning framework for enhanced resolution (*Figure 2—figure supplement 2D,E*; *Weigert et al., 2018*; *Wu et al., 2016*; *Wu et al., 2021*).

Using these methods, we observed that the AIB neurites enter the nerve ring during the early embryonic elongation phase, ~ 400 min post fertilization (m.p.f). The two AIB neurites then circumnavigate the nerve ring at opposite sides of the neuropil - both AIBL and AIBR project dorsally along the posterior neighborhood, on the left and right-hand sides of the worm, respectively (*Figure 2A and B*). Simultaneous outgrowth of AIBL and AIBR neurons in the posterior neighborhood results in their neurites circumnavigating the ring and meeting at the dorsal midline of the nerve ring (*Figure 2C*). Therefore, proper placement of the proximal segment of the AIB neurite in the posterior neighborhood occurs by AIB outgrowth along neurons in this neighborhood (*Figure 2A–F*).

After meeting at the dorsal midline, instead of making a shift to the anterior neighborhood (as expected from the adult AIB neurite morphology – see *Figure 1M and N*), the AIB neurites, surprisingly, continue growing along the posterior neighborhood (*Figure 2C and D*; 480 m.p.f.). At approximately 505 m.p.f., each AIB neurite separates from the posterior neighborhood, starting at its growth cone, by growing tangentially to the posterior neighborhood (the posterior neighborhood is marked in *Figure 2A–G* by its lateral counterpart, that is, the other AIB, also see *Figure 2—figure supplement 2I,J*). The departure of the AIB growth cone occurs due to the AIB neurite growing in a straight path trajectory instead of following the bending nerve ring arc (*Figure 2—figure supplement 2I,J*). Because it has been documented that axons tend to 'grow straight' on surfaces lacking adhesive forces that instruct turning (*Katz, 1985*), we hypothesize that the observed exit (via 'straight outgrowth') could result from decreased adhesion to the posterior neighborhood (*Figure 2—figure supplement 2I,J*).

As it grows tangentially to the posterior neighborhood, the AIB neurite cuts orthogonally through the nerve ring and toward the anterior neighborhood (*Figure 2—figure supplement 2I,J*). Upon intersecting the anterior neighborhood, the AIB neurite reengages with the arc of the nerve ring. At this developmental stage (*Figure 2I*), only 3.9 % of the AIB distal neurite is placed in the anterior neighborhood, with the remainder still being positioned in the posterior neighborhood and between neighborhoods. Following this, we observed a repositioning of the AIB neurite, but not via expected tip-directed fasciculation. Instead, the entire shaft of the distal AIB neurite was peeled away from the posterior neighborhood and repositioned onto the anterior neighborhood, starting from the tip of the neurite and progressively 'zippering' in a retrograde fashion towards the cell body (*Figure 2J and K*; the overlap of the AIB neurite with the anterior neighborhood increased from 3.9 % at 515 m.p.f. to 30.4 % at 530 m.p.f. and 71.7 % at 545 m.p.f.). Retrograde zippering stopped at the dorsal midline of the nerve ring (~545 m.p.f.), resulting in the AIB architecture observed in postembryonic larval and adult stages (*Figure 2L*). The progressive zippering of the AIB neurite onto the anterior neighborhood occurs concurrently with its separation from the posterior neighborhood (*Figure 2M*), a converse process which we refer to as 'unzippering'. The in vivo developmental dynamics of AIB repositioning, via retrograde zippering onto the anterior neighborhood, are reminiscent of dynamics observed in cultures of vertebrate neurons in which biophysical forces drive 'zippering' of neurite shafts, and the bundling of neurons (*Smít et al., 2017*). This mechanism is distinct from neurite bundling directed

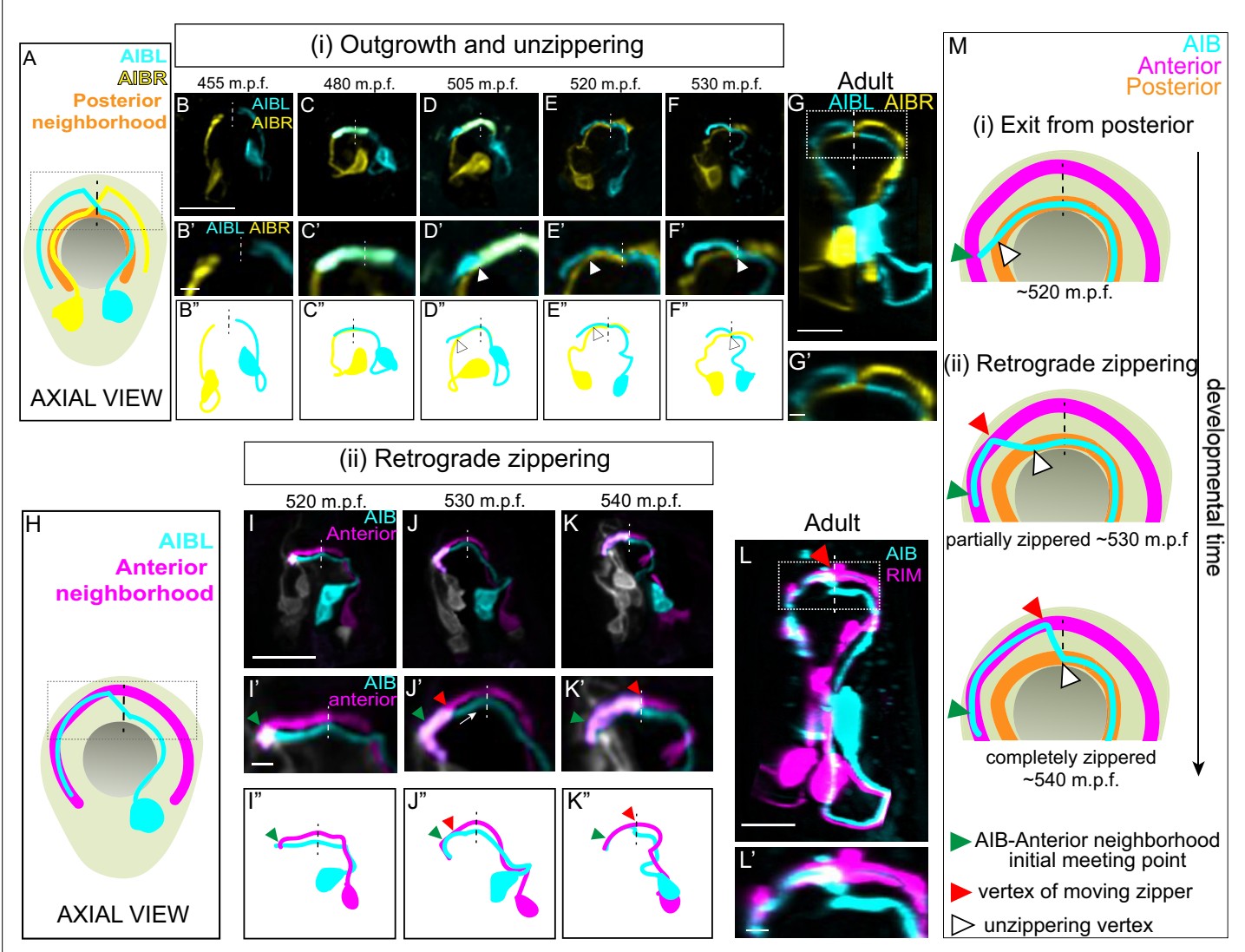

**Figure 2.** A retrograde zippering mechanism positions the AIB neurites in the anterior neighborhood during embryonic development. **A**, Schematic of axial view of the AIB neuron pair: AIBL (cyan) and AIBR (yellow) in the context of the nerve ring (light neon) and the pharynx (grey), with posterior neighborhood labeled (orange) and the dashed line representing the dorsal midline where the AIB chiasm is present in adults (see *Figure 1*). Dotted box represents region in **B'-F'**, and dotted box in **G**. **B,F**, Time-lapse showing initial placement of AIBL and AIBR in the posterior neighborhood and their subsequent separation from this neighborhood. Images are deconvolved diSPIM maximum intensity projections obtained from developing embryos. Neurons were individually pseudocolored to distinguish them (see Methods). The dorsal half of the nerve ring (dotted box in **A**) are magnified in **B'-F'**. **B"-F"** are schematic diagrams representing the images in **B-F**. Dashed vertical lines midline. Note in (**B, B', B"**), the AIBL and AIBR neurites approaching the dorsal midlinerepresent the dorsal in the posterior neighborhood. In (**C, C', C"**), AIBL and AIBR have met at the dorsal midline and continue growing along each other, past the midline. The latter part of the neurite, past the midline, becomes the future distal neurite. (**D, D', D"**) shows the tip of the AIBL future distal neurite moving away from the posterior neighborhood and its counterpart, AIBR. The arrowhead indicates the point of separation of the AIBL distal neurite and the AIBR proximal neurite. (**E, E', E"**) shows further separation of the two neurites and by (**F, F', F"**), they have completely separated. The arrowheads in (**E, E', E"**) and (**F, F', F"**) also indicate the junction between the separating AIBL distal neurite and the AIBR proximal neurite. A similar sequence of events is visualized at higher spatial resolution in *Figure 2—figure supplement 1* using triple-view line scanning confocal microscopy (*Figure 2—figure supplement 1*). **G, G'**, Confocal micrograph of a postembryonic L4 animal (axial view) showing the relationship between AIBL and AIBR. The region in the box represents the dorsal part of the nerve ring, magnified in **G'**. **H**, Axial view schematic of one AIB neuron (cyan) in the context of the anterior neighborhood marker, the RIM neuron (magenta), the nerve ring (light neon) and the pharynx (grey). **I-K**, Time-lapse showing placement of the AIB neurite (cyan) relative to the anterior neighborhood (magenta). As in **B-F**, images are deconvolved diSPIM maximum intensity projections and the neurons were pseudocolored. The dorsal half of the nerve ring (dotted box in **H**) are magnified in **I'-K'**. Dashed line indicates dorsal midline (where the shift, or chiasm, in the adult is positioned, see *Figure 1*). **I"-K"** are schematic diagrams representing the images in **I-K**. Note in (**I, I', I"**), the tip of the AIB neurite encounters the RIM neurite in the anterior neighborhood (green arrowhead). In (**J, J', J"**), the AIB distal neurite has partially aligned along the RIM neurites. The green arrowhead now indicates point of initial encounter of the two neurites (same as in **I'**), and

*Figure 2 continued on next page*

*Figure 2 continued*

the red arrowhead indicates the retrograde zippering event bringing the AIB and RIM neurons together in the anterior neighborhood. In (**K, K', K"**) the two neurites have zippered up to the dorsal midline. Arrow in **J'** indicates direction of zippering. **L**, Confocal micrograph of a postembryonic L4 animal in axial view showing the final position of AIB with respect to the anterior neighborhood. The same image as *Figure 1I* was used here for reference. The region in the dotted box represents dorsal part of the nerve ring, magnified in (**L'**). **M**, Schematic highlights the steps by which the AIB distal neurite is repositioned to a new neighborhood – (i) exit from the posterior neighborhood and (ii) retrograde zippering onto the anterior neighborhood with intermediate partially zippered states and completely zippered states. Scale bar = 10 μm for **B-G** and **I-L**. Scale bar = 2 μm for **B'-G'** and **I'-L'** Times are in m.p.f. (minutes post fertilization).

The online version of this article includes the following video and figure supplement(s) for figure 2:

**Figure supplement 1.** Images of AIBL and AIBR unzippering with and without pseudocoloring.

**Figure supplement 2.** Strategies for labeling and visualizing AIB outgrowth dynamics in early embryos.

**Figure 2—video 1.** Selective AIB labeling by Zif-1/ZF1-mediated degradation, related to Figure 2 and Figure 2—figure supplement 2.
https://elifesciences.org/articles/71171/figures#fig2video1

**Figure 2—video 2.** diSPIM single-view z-stacks for Figure 2B.
https://elifesciences.org/articles/71171/figures#fig2video2

**Figure 2—video 3.** diSPIM single-view z-stack for Figure 2C.
https://elifesciences.org/articles/71171/figures#fig2video3

**Figure 2—video 4.** diSPIM single-view z-stack for Figure 2D.
https://elifesciences.org/articles/71171/figures#fig2video4

**Figure 2—video 5.** diSPIM single-view z-stack for Figure 2E.
https://elifesciences.org/articles/71171/figures#fig2video5

**Figure 2—video 6.** diSPIM single-view z-stack for Figure 2F.
https://elifesciences.org/articles/71171/figures#fig2video6

**Figure 2—video 7.** 3D projection of deconvolved, fused diSPIM image for timepoint 455 m.p.f. (Figure 2B).
https://elifesciences.org/articles/71171/figures#fig2video7

**Figure 2—video 8.** 3D projection of deconvolved, fused diSPIM image for timepoint 480 m.p.f. (Figure 2C).
https://elifesciences.org/articles/71171/figures#fig2video8

**Figure 2—video 9.** 3D projection of deconvolved, fused diSPIM image for timepoint 505 m.p.f. (Figure 2D).
https://elifesciences.org/articles/71171/figures#fig2video9

**Figure 2—video 10.** 3D projection of deconvolved, fused diSPIM image for timepoint 520 m.p.f. (Figure 2E).
https://elifesciences.org/articles/71171/figures#fig2video10

**Figure 2—video 11.** 3D projection of deconvolved, fused diSPIM image for timepoint 530 m.p.f. (Figure 2).
https://elifesciences.org/articles/71171/figures#fig2video11

by anterograde migration of neurite tips (*Bak and Fraser, 2003*), and retrograde zippering, until this study, had not been documented during development and in vivo.

## Biophysical modeling of AIB developmental dynamics is consistent with differential adhesion leading to retrograde zippering

Dynamics of neurite shaft zippering have been previously documented (*Barry et al., 2010*; *Voyiadjis et al., 2011*), modeled in tissue culture cells (*Smít et al., 2017*), and described as resulting from two main forces: neurite-neurite adhesion (represented as 'S') and mechanical tension (represented as 'T'). To better understand the mechanisms that act in vivo during AIB neurite placement, we analyzed AIB developmental dynamics in the context of these known forces that affect neurite zippering. In each neighborhood, the developing AIB neurite experiences two forces: (i) adhesion to neurons in that neighborhood and (ii) tension due to mechanical stretch. As the neurite zippers and unzippers, it has a velocity in the anterior neighborhood (a zippering velocity, $v_{zip}$) and a velocity in the posterior neighborhood (an unzippering velocity, $v_{unzip}$) (*Figure 3*, Appendix 1 and *Appendix 1—figure 1*). These velocities are related to the forces on the neurite by the following equation:

$$v_{zip} + v_{unzip} = \frac{(S_{anterior} - S_{posterior})}{\eta} - \frac{\Delta T}{\eta}\left(1 - cos\theta\right)$$

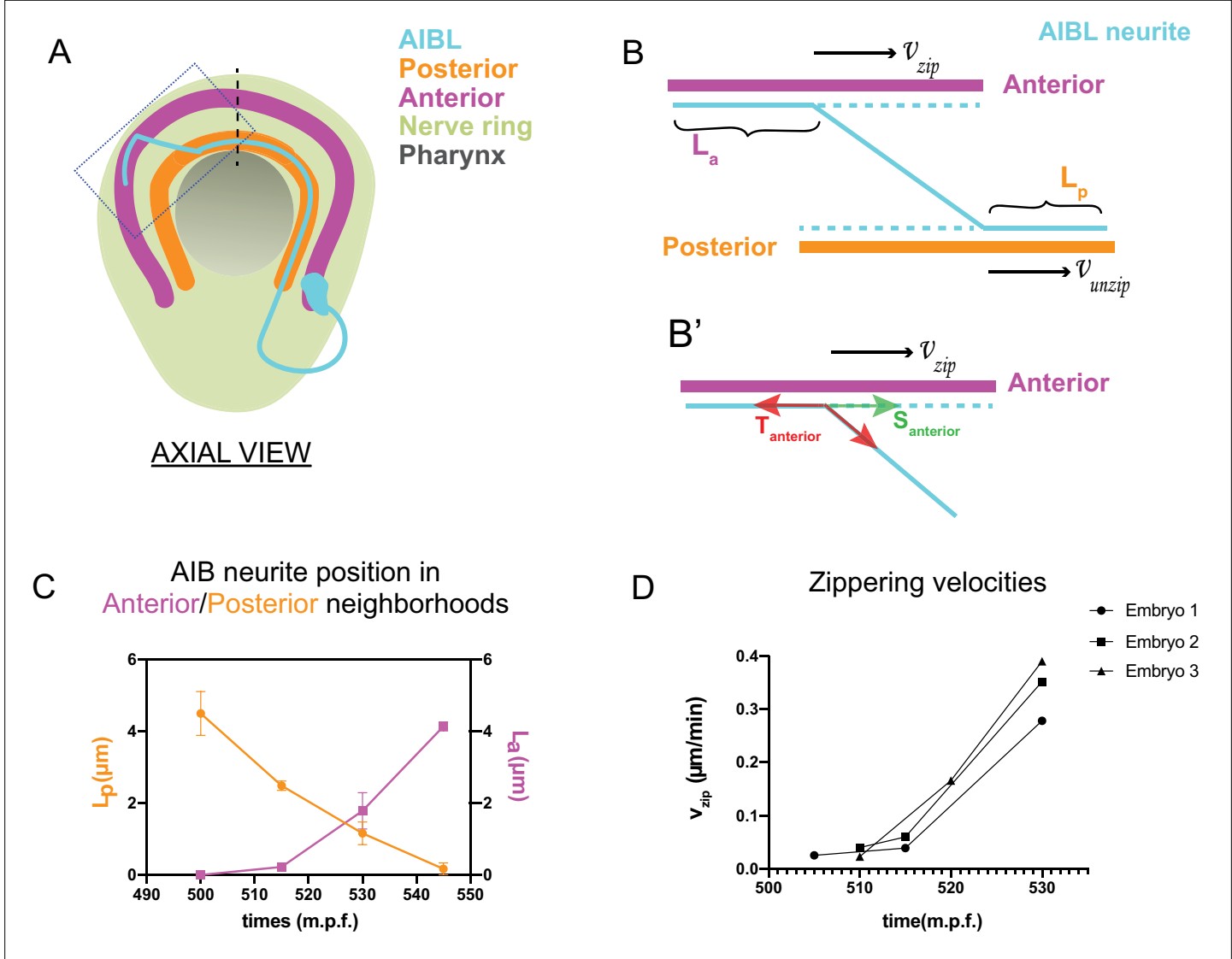

**Figure 3.** Biophysical modeling of AIB developmental dynamics is consistent with differential adhesion leading to retrograde zippering. (**A**) Axial view schematic of a single AIB neuron during transition of its neurite between the posterior (orange) and anterior (magenta) neighborhoods. (**B,B'**) Magnified schematic of dotted inset in (**A**) showing the AIB neurite (cyan) during its transition from the posterior to the anterior neighborhood. The lengths of the neurite positioned in the posterior and anterior neighborhoods are denoted by $L_p$ and $L_a$, respectively. The velocity with which the AIB neurite zippers onto the anterior neighborhood is denoted by $v_{zip}$, and the velocity with which it unzippers from the posterior neighborhood is denoted by $v_{unzip}$. At the junction between the neurite and the two neighborhoods, that is at the zippering and unzippering forks, tension and adhesion forces act on the neurite (see B', Appendix 1 and **Appendix 1—figure 1**). B', Schematic of AIB neurite zippering to the anterior neighborhood. Adhesion $S_{anterior}$ acts in the direction of zippering (and therefore in the direction of the zippering velocity $v_{zip}$) and favors zippering. Tension $T_{anterior}$ acts in the opposite direction, disfavoring zippering. (**C**) Plot of position vs. time of the AIB neurite in both neighborhoods in synchronized embryos at the indicated timepoints on the x-axis ( ± 5 mins). Plot shows mean of $L_p$ (n = 4) and $L_a$ (n = 3) values at different timepoints. Note zippering from the anterior neighborhood and unzippering from the posterior neighborhood take place in the same time window and are inversely related (between 500–545 m.p.f.). Quantifications were done from three embryos for each of $L_a$ and $L_p$. See **Figure 3—figure supplement 1** for the individual $L_p$ and $L_a$ values at each timepoint. (**D**) Plot of zippering velocities vs time (n = 3) for the indicated timepoints on the x-axis ( ± 5 mins). Note a tenfold increase in velocity mid-way through zippering (530 m.p.f.) m.p.f. = minutes post fertilization. Error bars represent standard error of the mean (S.E.M.), The three embryo datasets used for measuring $L_a$ values in (**C**) were used to calculate zippering velocities. For **C** and **D**, n represents the number of AIB neurites quantified.

The online version of this article includes the following figure supplement(s) for figure 3:

**Figure supplement 1.** Individual data points from **Figure 3C**.

where $v_{zip}$ = zippering velocity, $v_{unzip}$ = unzippering velocity, $S_{anterior} - S_{posterior}$ = difference between adhesive forces in the two neighborhoods, $T = T_{anterior} - T_{posterior}$ = difference between tension acting on the AIB neurite in the two neighborhoods, $\eta$ = friction constant (see Appendix 1 and *Appendix 1—figure 1*) and $\theta$ = angle of the AIB neurite to the neighborhoods (*Figure 3*, Appendix 1 and *Appendix 1—figure 1*). Since the above biophysical equation defines the relationship between velocities and forces, we measured the velocities of the neurite from our time-lapse images to make predictions about the forces on the neurite.

Time lapse images and measurements of the developmental dynamics showed that zippering and unzippering takes place concurrently: zippering on to the anterior neighborhood and unzippering from the posterior one (*Figure 3C*). Between 505 and 545 m.p.f., the average length of the AIB neurite that is placed in the anterior neighborhood (4.49 µm) by retrograde zippering is similar to the length that is unzippered from the posterior neighborhood (4.13 µm). Assuming, based on previous studies (*Smít et al., 2017*), that the tension forces are uniformly distributed along the neurite (and therefore $\Delta T = T_{anterior} - T_{posterior} = 0$) zippering and unzippering velocities arise from a difference in adhesion ($S_{anterior} - S_{posterior} > 0$) (see Appendix 1 and *Appendix 1—figure 1*).

Measurements of in vivo zippering velocities (*Figure 3D*) support this hypothesis. Examination of our time-lapse images revealed that AIB neurite zippering onto the distal neighborhood takes place at higher velocities at later timepoints (with mean zippering velocity increasing from 0.09 µm/min at 515 min to 0.34 µm/min at 530 min) (*Figure 3D*). This increased velocity, or acceleration, is a hallmark of force imbalance and consistent with a net increase in adhesive forces in the anterior neighborhood during the period in which zippering takes place. We note that retrograde zippering comes to a stop precisely at the dorsal midline, likely owing to the adhesion and tension forces on the neurite in the two neighborhoods balancing out at this point.

Together, the developmental dynamics observed for AIB neurite placement are consistent with relative changes in adhesive forces between the neighborhoods. This suggests that dynamic mechanisms resulting in differential adhesion might govern AIB neurite repositioning by a process similar to affinity-based sorting of cells within homogenous tissues (*Steinberg, 1963*; *Steinberg, 1970*). We show that differential adhesion across nerve ring bundles result in neurite placement by a zipper-like mechanism (*Barry et al., 2010*; *Roberts and Taylor, 1982*; *Voyiadjis et al., 2011*), distinct from the classical paradigm of chemical attraction of the growing neurite tip to pre-existing nerve bundles or guidepost cells (*Plachez and Richards, 2005*; *Sabry et al., 1991*).

## SYG-1 and SYG-2 regulate precise placement of the AIB neurite to the anterior neighborhood

To identify the molecular mechanism underpinning differential adhesion for AIB neurodevelopment, we performed forward and reverse genetic screens (see Materials and methods). We discovered that loss-of-function mutant alleles of *syg-1* and *syg-2*, which encode a pair of interacting Ig family cell adhesion molecules (IgCAMs), display significant defects in the placement of the AIB neurite. In wild type animals, we reproducibly observed complete overlap between the AIB distal neurite and neurons in the anterior neighborhood (*Figure 4A–D*), consistent with EM characterizations (*Figure 1—figure supplement 2A,B*). In contrast, 76.3 % of *syg-1(ky652)* animals and 60 % of *syg-2(ky671)* animals (compared to 1.8 % of wild-type animals) showed regions of AIB detachment from neurons specifically in the anterior neighborhood (*Figure 4E–L*; we note we did not detect defects in general morphology of the nerve ring, in the length of the AIB distal neurite, or in position of the AIB neurite in the posterior neighborhood for these mutants, *Figure 4—figure supplement 1*). In the *syg-1(ky652)* and *syg-2(ky671)* animals that exhibit defects in AIB neurite placement, we found that 20.9 ± 3.9 and 18.6% ± 4.0% (respectively) of the neurite segment in the anterior neighborhood is detached from the neighborhood (*Figure 4M*). Our findings indicate that SYG-1 and SYG-2 are required for correct placement of AIB, specifically to the anterior neighborhood.

The IgCAMs SYG-1 and SYG-2 are a receptor-ligand pair that has been best characterized in the context of regulation of synaptogenesis in the *C. elegans* egg-laying circuit (*Shen and Bargmann, 2003*; *Shen et al., 2004*). SYG-1 (Rst and Kirre in *Drosophila* and Kirrel1/2/3 in mammals) and SYG-2 (Sns and Hibris in *Drosophila*, and Nephrin in mammals) orthologs also act as multipurpose adhesion molecules in varying conserved developmental contexts (*Bao and Cagan, 2005*; *Bao et al., 2010*; *Chao and Shen, 2008*; *Garg et al., 2007*; *Neumann-Haefelin et al., 2010*; *Ozkan et al., 2014*;

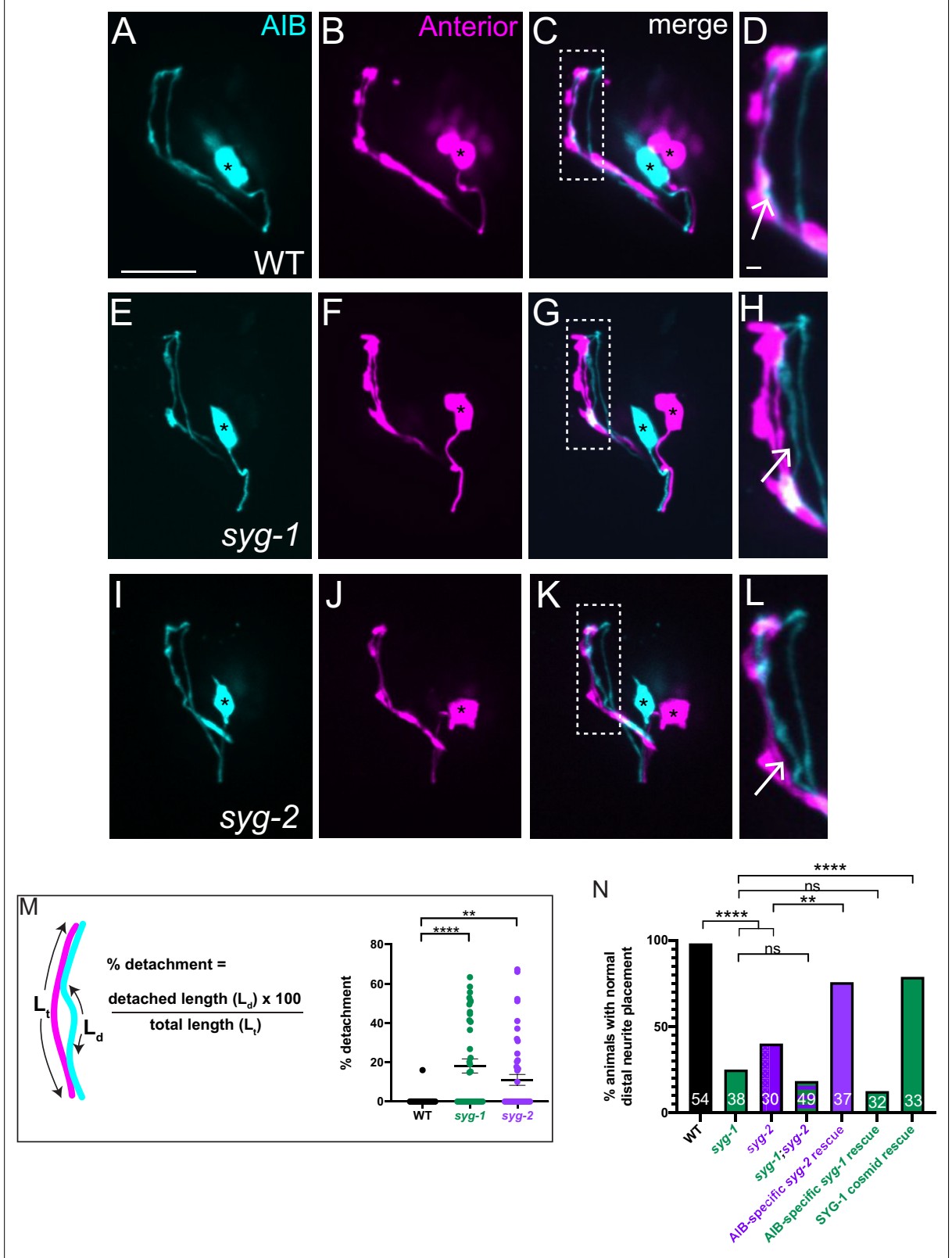

**Figure 4.** SYG-1 and SYG-2 are required for precise placement of the AIB neurite in the anterior neighborhood. (**A-D**) Representative confocal images of AIB (**A**) and RIM neurons (**B**) which mark the anterior neighborhood, in a wild-type animal. (**C**) is a merge of **A** and **B**. The dashed box represents the region of contact of AIB with the anterior neighborhood, magnified in (**D**). The AIB distal neurite colocalizes extensively with the anterior neighborhood in wild-type animals (Arrow in **D** and *Figure 1—figure supplement 2A,B*). Cell bodies are marked with an asterisk. (**E–L**) As **A–D** but in the *syg-1(ky652)*

*Figure 4 continued on next page*

*Figure 4 continued*

(**E–H**) and *syg-2(ky671)* (**I–L**) mutant background. Note the gaps between the AIB distal neurite and the RIM neurites (**H,L**, arrows), indicating loss of contact between the AIB and the anterior neighborhood in these mutants. (**M**) Schematic and scatter plot of quantifications of the loss of contacts between AIB and the anterior neighborhood for wild type (n = 42), *syg-1(ky652)* mutant (n = 40) and *syg-2(ky671)* animals (n = 49). 'n' represents the number of AIB neurites quantified from 21, 20 and 25 animals, respectively. The extent of detachment of the AIB distal neurite, and hence its deviation from the RIM neighborhood, was quantified using the indicated formula (see also Materials and methods). Error bars indicate standard error of the mean (S.E.M.). ****p < 0.0001, **p = 0.0095 (one-way ANOVA with Dunnett's multiple comparisons test). n represents the number of AIB neurites quantified. Estimated effect size, d = 1.087 for WT vs. *syg-1(ky652)* and 0.775 for WT vs. *syg-2(ky671)*. For neurites that do not show visible detachment, the precent detachment values = 0 and therefore these datapoints lie on the x-axis. The mean percent deviations include neurites with 0 percent detachment. (**N**) Quantification of the penetrance of the AIB neurite placement defect as the percentage of animals with normal AIB distal neurite placement in WT, *syg-1(ky652)*, *syg-2(ky671)*, *syg-1(ky652);syg-2(ky671)* double mutant, *inx-1p:syg-2* rescue, *inx-1p:syg-1* rescue and SYG-1 cosmid rescue (also see *Figure 4—figure supplement 1G-I'*). *inx-1p* is a cell-specific promoter driving expression in AIB (*Altun and Chen, 2008*). The green and purple bars represent *syg-1(ky652)* and *syg-2(ky671)* mutant backgrounds respectively. Numbers on bars represent number of animals examined. ****p < 0.0001 by two-sided Fisher's exact test between WT and *syg-1(ky652)*, between WT and *syg-2(ky671)*, and between *syg-1(ky652)* and SYG-1 cosmid rescue, and **p = 0.0055 between *syg-2(ky671) and inx-1p:syg-2* rescue. There is no significant difference (abbreviated by n.s.) in penetrance between the *syg-1(ky652)* and *syg-1(ky652);syg-2(ky671)* (p = 0.6000) populations and between *syg-1(ky652)* and the *inx-1p:syg-1* animals (p = 0.3558). Scale bar = 10 μm, applies to (**A–L**).

The online version of this article includes the following source data and figure supplement(s) for figure 4:

**Source data 1.** Number of animals of each genotype (in the bar graph in *Figure 4N*) displaying normal vs aberrant distal neurite placement.

**Figure supplement 1.** SYG-1 and SYG-2 regulate placement of the AIB neurite specifically in the anterior neighborhood.

*Oztokatli et al., 2012*; *Serizawa et al., 2006*; *Shen and Bargmann, 2003*; *Shen et al., 2004*; *Strünkelnberg et al., 2001*). In most of the characterized in vivo contexts, SYG-1 has been shown to act heterophilically with SYG-2 (*Dworak et al., 2001*; *Ozkan et al., 2014*; *Shen et al., 2004*). Consistent with SYG-1 and SYG-2 acting jointly for precise placement of the AIB neurite in vivo, we observed that a double mutant of the *syg-1(ky652)* and *syg-2 (ky671)* loss-of-function alleles did not enhance the AIB distal neurite placement defects as compared to either single mutant (*Figure 4N*).

To determine the site of action of these two molecules, we expressed them cell-specifically in varying tissues. We observed that SYG-2 expression in AIB was sufficient to rescue the AIB distal neurite placement defects in the *syg-2(ky671)* mutants, suggesting that SYG-2 acts cell autonomously in AIB. While expression of wild-type SYG-1 (via a cosmid) rescued AIB neurite placement onto the anterior neighborhood, expression of SYG-1 using an AIB cell-specific promoter did not (*Figure 4N*), consistent with SYG-1 regulating AIB neurite placement cell non-autonomously.

## Increased local expression of SYG-1 in the anterior neighborhood coincides with zippering of the AIB neurite onto this neighborhood

To understand how SYG-1 coordinates placement of the AIB neurite, we examined the expression of transcriptional and translational reporters of SYG-1 in the nerve ring of wild type animals. In postembryonic, larva-stage animals (L3 and L4), we observed robust expression of the *syg-1* transcriptional reporter in a banded pattern in ~20 neurons present in the AIB posterior and anterior neighborhoods, with specific enrichment in the anterior neighborhood (*Figure 5A–E*). The SYG-1 translational reporter, which allowed us to look at SYG-1 protein accumulation, also showed a similar expression pattern (*Figure 5F–I*). To understand how SYG-1 regulates placement of the AIB neurite during development, we examined spatiotemporal dynamics of expression of SYG-1 during embryogenesis at the time of AIB neurite placement (400–550 m.p.f.) (*Figure 2*), using both the transcriptional and translational *syg-1* reporters.

Prior to 470 m.p.f., *syg-1* reporter expression in the nerve ring was primarily restricted to a single band corresponding to the AIB posterior neighborhood (*Figure 5K, O and O'*). This coincides with periods of outgrowth and placement of the AIB neurons in the posterior neighborhood. However, over the subsequent three hours of embryogenesis (470–650 m.p.f.), SYG-1 expression levels progressively increase in the anterior neighborhood while decreasing in the posterior neighborhood (*Figure 5L–R'*, *Figure 5—figure supplement 1*, *Figure 5—video 1*). The change in expression levels of SYG-1 across neighborhoods coincides with the relocation of the AIB neurite, from the posterior to the anterior neighborhood via retrograde zippering (*Figure 5S*). To identify which neurons in the nerve ring express SYG-1, we performed single-cell lineaging (*Murray et al., 2006*) of the neurons expressing the *syg-1* transcriptional reporter at approximately 430 m.p.f. (*Figure 5—figure supplement 2A-C*,

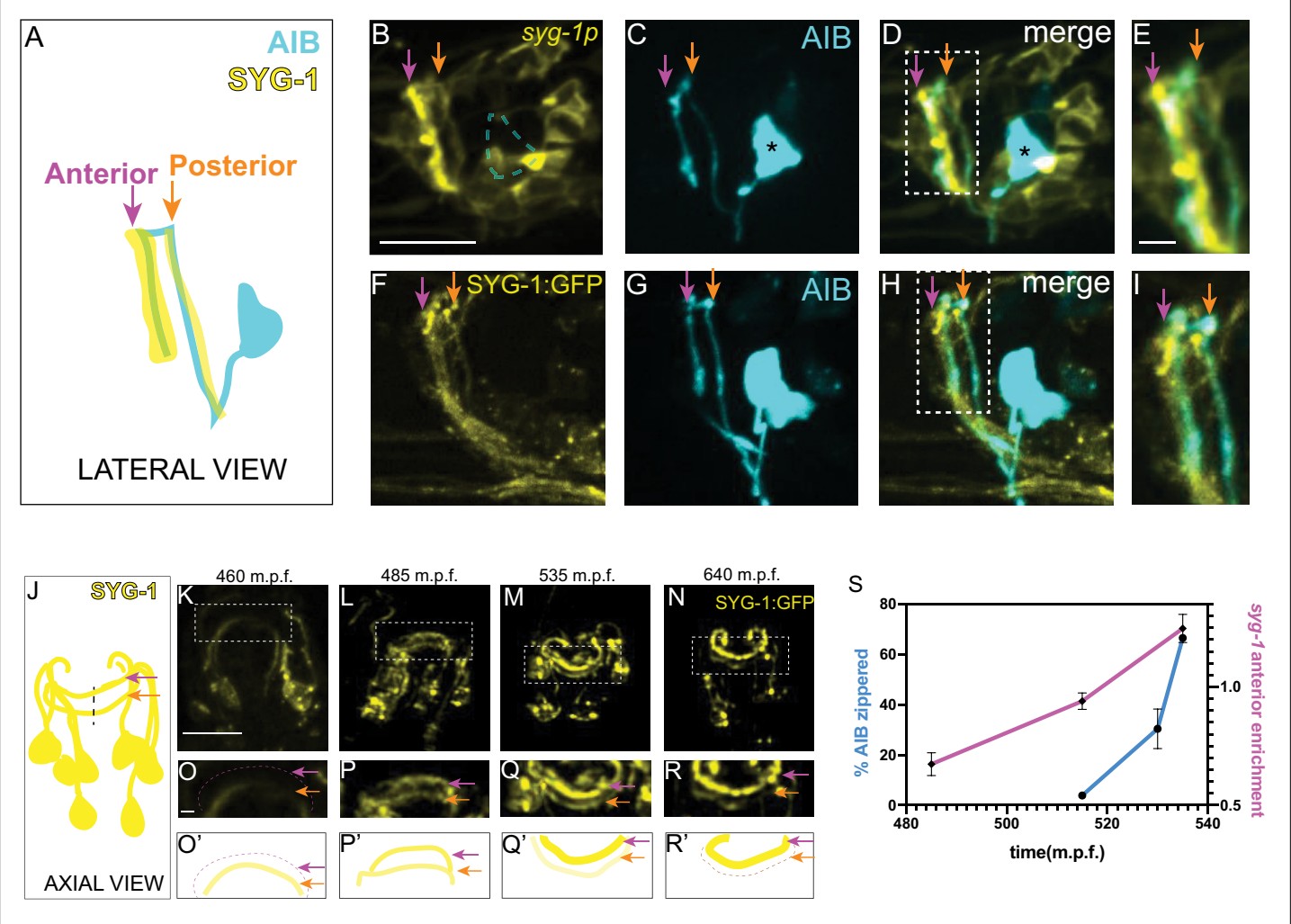

**Figure 5.** Increased local expression of SYG-1 in the anterior neighborhood coincides with zippering of the AIB neurite onto this neighborhood. (**A-E**) Schematic (**A**) and representative confocal image of a wild-type animal co-expressing (**B**) a membrane-targeted *syg-1* transcriptional reporter (see Materials and methods, *Schwarz et al., 2009*) and (**C**) cytoplasmic AIB reporter. Merged image in (**D**). Since the *syg-1* reporter is membrane-targeted, it labels cell body outlines and neurites (**B, D**). The dashed box or inset in (**D**) represents the region of overlap between AIB and *syg-1*-expressing neurites, magnified in (**E**). Note that the *syg-1* reporter shows two bands of expression in the nerve ring (arrows in **B** and **D**) which coincide with the posterior and anterior AIB neighborhoods (orange and magenta arrows). Note also that there is no membrane outline corresponding to the AIB cell body (**B**) we drew a dashed silhouette of the AIB cell body position as determined in (**C**). Asterisk indicates cell body. (**F–I**) As **B–E**, but with a translational SYG-1 reporter. Note the SYG-1 protein shows a similar expression pattern. (**J–N**) Schematic (**J**) and time-lapse images (**K–N**) of SYG-1 translational reporter expression during embryogenesis (460–640 m.p.f.). Images are deconvolved diSPIM maximum intensity projections. The dashed boxes represent the dorsal half of the nerve ring and are magnified in O-R. O'-R' are schematic diagrams representing the images in (**O–R**). In (**K, O, O'**), SYG-1 expression is primarily visible in a single band containing amphid neurites and corresponding to the AIB posterior neighborhood. The magenta dashed line and magenta arrows point to the anterior neighborhood and the orange arrow, to the posterior neighborhood. By 535 m.p.f. (**L, P, P'**), SYG-1 expression is visible in both the anterior and posterior neighborhoods. In subsequent timepoints (**M, Q, Q', N, R, R'**), SYG-1 expression increases in the anterior neighborhood and decreases in the posterior neighborhood, coincident with AIB developmental events that enable its transition from the posterior to the anterior neighborhood (*Figure 2B–K*). The *syg-1* transcriptional reporter shows a similar expression pattern throughout development (*Figure 5— figure supplement 1*). (**S**) Plot showing relative enrichment of the *syg-1* transcriptional reporter in the anterior neighborhood over time (magenta) overlaid with plot showing percentage of the relocating AIB distal neurite that has zippered onto the anterior neighborhood (blue). Relative enrichment in the anterior neighborhood is defined as the ratio of mean intensity of the *syg-1* reporter in the band corresponding to the AIB anterior neighborhood, as compared to that in the posterior neighborhood (see Materials and methods). This value is calculated starting at a timepoint when *syg-1* reporter expression becomes visible in the anterior neighborhood and averaged for four embryos. The relative enrichment values plotted represent values calculated at the indicated developmental times on the x-axis ( ± 10 mins). The reported values of '% AIB zippered' are averaged across the three independent embryo datasets used for the plots in *Figure 3*. Note similar SYG-1 expression dynamics to zippering dynamics in AIB. Error bars represent

*Figure 5 continued on next page*

Figure 5 continued

standard error of the mean (S.E.M.). See **Figure 5—figure supplement 5** for the individual values of *syg-1* anterior enrichment and '% AIB zippered'. Scale bar = 10 µm, applies to **B–D**, (**F–H**) and **K–N**. Scale bar = 2 µm in **E, I** and **O–R**. Times are in m.p.f. (minutes post fertilization).

The online version of this article includes the following video, source data, and figure supplement(s) for figure 5:

**Figure supplement 1.** Spatiotemporal regulation of *syg-1* transcriptional reporter expression during embryogenesis.

**Figure supplement 2.** Neurons expressing SYG-1 in the embryonic nerve ring.

**Figure supplement 3.** SYG-1 is expressed in the anterior neighborhood RIM neurons.

**Figure supplement 4.** The RIM neurons regulate AIB distal neurite placement.

**Figure supplement 4—source data 1.** Number of animals of each genotype (in the bar graph in **Figure 5—figure supplement 4N**) displaying normal vs aberrant distal neurite placement.

**Figure supplement 5.** Individual data points from **Figure 5S**.

**Figure supplement 6.** Localization of SYG-2 puncta to the AIB distal neurite.

**Figure 5—video 1.** Layered expression of SYG-1 in the nerve ring in embryos, related to Figure 5 and Figure 5—figure supplement 1.
https://elifesciences.org/articles/71171/figures#fig5video1

**Figure 5—video 2.** Identification of SYG-1-expressing neurons by lineage tracking, related to Figure 5—figure supplement 2.
https://elifesciences.org/articles/71171/figures#fig5video2

**Figure 5—video 2**). The six neurons in the anterior neighborhood, and 10 neurons in the posterior neighborhood which we identified (**Figure 5—figure supplement 2C**), were consistent with the identity of SYG-1 expressing neurons from embryonic transcriptomics data (**Packer et al., 2019**). Both our data, and embryonic transcriptomics data, reveal dynamic changes in the expression levels of SYG-1 in these neurons (**Figure 5—figure supplement 2**). The transcriptomic studies also demonstrate a ten-fold increase in SYG-2 transcript levels in AIB at the time in which the AIB neurite transitions between neighborhoods (and consistent with our findings that SYG-2 acts cell autonomously in AIB). Together with the biophysical analyses, our data suggests that spatiotemporal changes in SYG-1 and SYG-2 expression might result in changes in forces that drive differential adhesion of AIB neurites via retrograde zippering of their axon shafts.

## Ectopic SYG-1 expression is sufficient to alter placement of the AIB distal neurite

To test whether coincident SYG-1 expression in the anterior neighborhood was responsible for repositioning of AIB to that neighborhood, we set to identify and manipulate the sources of SYG-1 expression. We found that increases of SYG-1 in the anterior neighborhood were caused by (i) ingrowth of SYG-1-expressing neurons into the anterior neighborhood and (ii) onset of *syg-1* expression in neurons of the anterior neighborhood (**Figure 5—figure supplement 1**). We observed strong and robust SYG-1 expression in the RIM neurons, as RIM grows into the anterior neighborhood, contributing to increased SYG-1 expression levels in this neighborhood. Since RIM is also one of the major fasciculation partners of AIB, we hypothesized that SYG-1 expression in RIM neurons contributes to AIB neurite placement (**Figure 5—figure supplement 3**). To test this hypothesis, we ablated RIM neurons. We observed that RIM ablations result in defects in AIB neurite placement which phenocopied those seen for *syg-1* loss-of-function mutants (**Figure 5—figure supplement 4**). We also observed that expression of SYG-1 specifically in RIM and RIC neurons in *syg-1(ky652)* mutants was sufficient to position the AIB distal neurite along these neurons (**Figure 5—figure supplement 4P-Q**).

If differences in SYG-1 expression level between the neighborhoods results in differential adhesion, and consequent relocation of the AIB distal neurite from the posterior to the anterior neighborhood, then purposefully altering these differences should predictably alter the position of the AIB neurite. We tested this hypothesis by inverting the adhesion differential through the overexpression of SYG-1 in the posterior neighborhood (see Materials and methods). Unlike wild type and *syg-1* mutants (**Figure 6A–F**, **Figure 6—figure supplement 1**), animals with ectopic *syg-1* expression in the posterior neighborhood displayed a gain-of-function phenotype, in which the AIB distal neurite remained partially positioned in the posterior neighborhood throughout postembryonic larval stages (**Figure 6G–J**, **Figure 6—figure supplement 1**). Importantly, these gain-of-function effects caused by ectopic expression of SYG-1 are not observed in a *syg-2(ky671)* mutant background (**Figure 6K**),

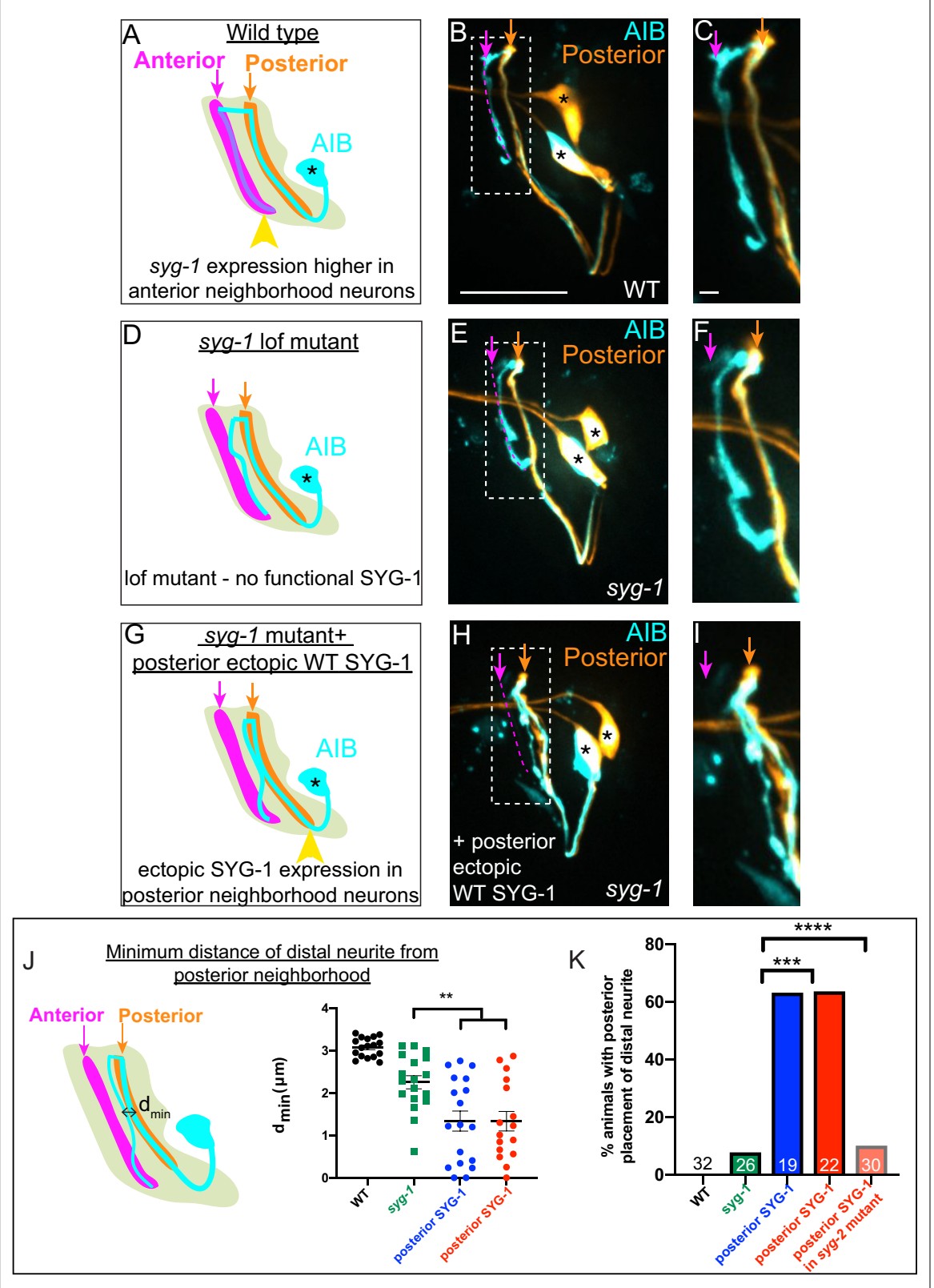

**Figure 6.** Ectopic *syg-1* expression is sufficient to alter placement of the AIB distal neurite. (**A**) Lateral view schematic of a wild-type AIB neuron (cyan) in the context of the posterior (orange) and anterior (magenta) neighborhoods, and the nerve ring (light neon). Higher SYG-1 endogenous expression in the anterior neighborhood represented by yellow arrowhead. (**B–C**) Confocal image of a wild type animal with AIB (labeled with cytoplasmic mCherry, in cyan) and the posterior neighborhood neurons AWC and ASE (labeled with cytoplasmic GFP, in orange). The dashed box represents the region

*Figure 6 continued on next page*

*Figure 6 continued*

of contact between AIB and the posterior neighborhood neurons, magnified in (**C**). Magenta dashed line represents the AIB anterior neighborhood. (**D–F**) As (**A–C**), but in the *syg-1(ky652)* lof (loss of function) mutant background. Note that the distal neurite is positioned away from the posterior neighborhood, as in wild type, although these animals display defects in fasciculation with the anterior neighborhood (as shown in *Figure 4*). (**G–I**) As (**D–F**), but with ectopic overexpression of SYG-1 in the posterior neighborhood neurons. In the schematic (**G**), expression of SYG-1 in the posterior neighborhood (achieved here using *nphp-4p*, also see *Figure 6—figure supplement 1*) is represented by a yellow arrowhead (as in (**A**), but here in posterior neighborhood). Note that the AIB distal neurite is now abnormally positioned in the posterior neighborhood in which SYG-1 was ectopically expressed (**H, I**). (**J**) Schematic (left) and scatter plot quantification (right) of minimum perpendicular distances ($d_{min}$, indicated by black double-headed arrow) between the AIB distal neurite and posterior neighborhood neurons in WT (in black, n = 17), *syg-1(ky652)* (in green, n = 18), and two *syg-1(ky652)* populations with SYG-1 overexpressed in two different sets of posterior neighborhood neurons via the use of *nphp-4p* and (in blue) *mgl-1bp* (in red) (n = 18 and n = 16 respectively). **p = 0.0056 and 0.0070, respectively (one-way ANOVA with Dunnett's multiple comparisons test). Effect size estimate, d = 1.075 and 1.140, respectively. Error bars indicate standard error of the mean (S.E.M.). n represents the number of AIB neurites quantified. Quantifications were done from nine animals each for WT, *syg-1(ky652)* and nphp-4p:syg-1; *syg-1(ky652)* and eight animals for mgl-1bp:syg-1; *syg-1(ky652)*. (**K**) Quantification of penetrance of the ectopic AIB neurite placement represented as the percentage of animals with the AIB distal neurite partially positioned in the posterior neighborhood in the WT, *syg-1(ky652)*, posterior SYG-1 overexpression strains (colors represent the same strains as in **J**) and a posterior SYG-1 overexpression strain in *syg-2(ky671)* background. Numbers on bars represent number of animals examined. ***p = 0.0002 for *syg-1(ky652)* and *nphp-4p* expressed SYG-1 and ****p < 0.0001 for *syg-1(ky652)* and *mgl-1bp* expressed SYG-1 by two-sided Fisher's exact test (also see *Figure 6—figure supplement 1*). Scale bar = 10 µm in **B**, **E** and **H** and 1 µm in **C**, **F**, and **I**. Cell body is marked with an asterisk.

The online version of this article includes the following source data and figure supplement(s) for figure 6:

**Source data 1.** Numbers of animals of each genotype (in *Figure 6K*) displaying ectopic distal neurite placement in the posterior neighborhood.

**Figure supplement 1.** A SYG-1-SYG-2 interaction underlies ectopic AIB neurite placement.

**Figure supplement 1—source data 1.** Numbers of animals of each genotype (in *Figure 6—figure supplement 1C*) displaying ectopic distal neurite placement in the posterior neighborhood.

consistent with SYG-2 expression in AIB being required for AIB's repositioning to the SYG-1 expressing layers. Our findings indicate that inverting the adhesion differential via enrichment of SYG-1 in the 'wrong' neighborhood predictably affects relocation of the AIB distal neurite in a way that is consistent with differential adhesion mechanisms.

We reasoned that if differential adhesion mechanisms were driving zippering of the AIB neurite during development, expression of the SYG-1 ectodomain would be sufficient to drive the ectopic interactions upon misexpression (*Chao and Shen, 2008*; *Galletta et al., 2004*; *Gerke et al., 2003*). Indeed, expression of the SYG-1 ectodomain in the posterior neighborhood resulted in gain-of-function phenotypes for AIB neurite placement, similar to those seen with misexpression of full-length SYG-1 (although penetrance of these effects was lower than that observed with full-length SYG-1). Consistent with the importance of adhesion-based mechanisms in the observed phenotypes, ectopic expression of the SYG-1 endodomain (which lacks the extracellular ectodomain necessary for interaction with SYG-2, see Materials and methods) in the posterior neighborhood did not result in mislocalization of AIB (*Figure 6—figure supplement 1*).

## AIB neurite placement by retrograde zippering, and presynaptic assembly, are coordinated during development

AIB displays a polarized distribution of pre- and postsynaptic specializations, and these specializations specifically localize to the neurite segments occupying the anterior and posterior neighborhoods, respectively. The placement of the AIB neurite in the anterior and posterior neighborhoods and its synaptic polarity underlies its role as a connector hub across layers (*Sabrin, 2019*; *Towlson et al., 2013*). To understand how the distribution of presynaptic specializations relates to the placement of the AIB neurite, we imaged the subcellular localization of presynaptic proteins RAB-3, CLA-1, and SYD-2 during AIB embryonic development. We observed that presynaptic proteins populate the AIB neurite starting from the tip toward the dorsal midline, in a retrograde pattern reminiscent of the retrograde zippering that places the AIB neurite in the anterior neighborhood (*Figure 7A–I*). The timing of formation of presynaptic sites suggested that that the process of synaptogenesis closely followed the retrograde zippering mechanisms of AIB repositioning (*Figure 7J and K*, *Figure 7— figure supplement 1*). Consistent with synaptogenesis occurring after retrograde zippering, we observed that a novel allele of *syd-2(ola341)* isolated from our screens exhibit synaptic defects, but do not display phenotypes in AIB neurite placement within the anterior neighborhood (*Figure 7—figure*

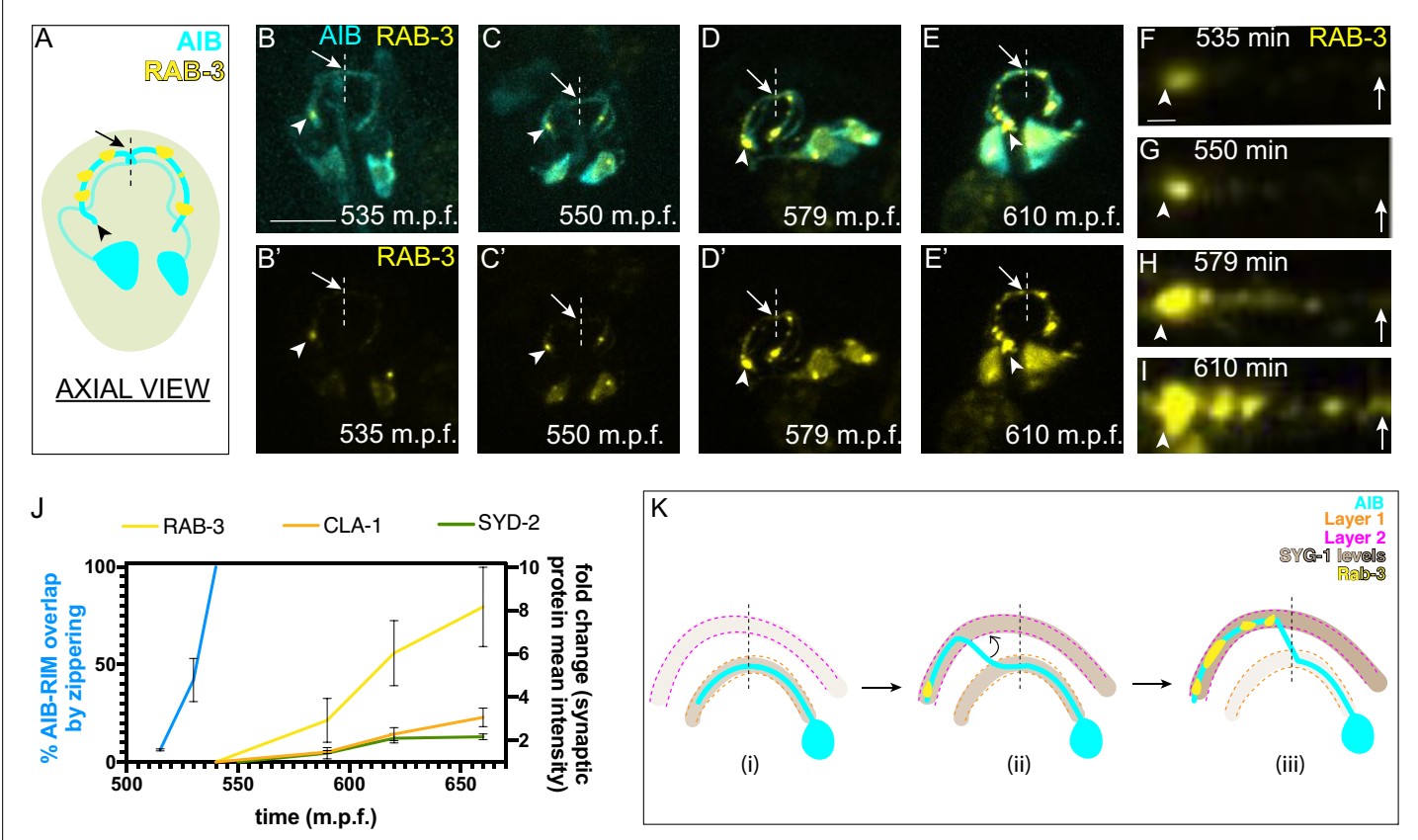

**Figure 7.** AIB neurite placement by retrograde zippering, and presynaptic assembly are coordinated during development. (**A**) Axial view schematic of the AIB neurons (cyan) with presynaptic protein RAB-3 (yellow) puncta along the distal neurite. Arrowhead indicates the tip of the distal neurite and arrow/dashed line indicate the dorsal midline. (**B–E**) Time-lapse imaging of RAB-3 localization in AIB during embryogenesis. (**B–E**) are merged diSPIM maximum intensity projections of AIB labeled with membrane-tagged mCherry (cyan) and AIB presynaptic sites labeled with GFP:RAB-3 (yellow), at different timepoints during embryogenesis. (**B'-E'**) represent the GFP:RAB-3 channel for images in **B–E**). Note in (**B, B'**) and (**C, C'**) that the RAB-3 signal in the neurite is localized exclusively near the neurite tip. As development progresses, there is more RAB-3 signal throughout the neurite from the tip up to the midline (in (**D, D'**) and (**E, E'**)). Therefore, RAB-3 becomes progressively enriched from the tip up to the midline during development, and the timing for this process correlates, with a slight delay, with the developmental timing of AIB zippering (*Figure 2I–K*). Arrowhead and arrow, as in (**A**), indicate the tip of the distal neurite and the region of the neurite near the dorsal midline (dashed vertical line) respectively. Scale bar = 10 μm applies (**B–E**) and (**B'-E'**). (**F–I**) Straightened distal neurites from AIB (corresponding to the region in (**B–E**) which is marked by the arrowhead (AIB tip) and arrows (dorsal midline)). Note presynaptic assembly, as imaged by RAB-3 accumulation, from the tip of the neurite towards the midline of AIB, reminiscent of the zippering event (*Figure 2*). Scale bar = 1 μm. (**J**) Plot showing average RAB-3, CLA-1 and SYD-2 intensities along the AIB distal neurite over time (yellow, orange and green, respectively) and percentage of the relocating AIB distal neurite that has zippered onto the anterior neighborhood (blue). See *Figure 7—figure supplement 1* for images of CLA-1 developmental dynamics in AIB. The intensities in the plot represent values calculated at the indicated developmental times on the x-axis ( ± 10 min). The reported values of '% AIB zippered' are averaged are the same as in *Figure 5S*. Note that RAB-3, CLA-1 and SYD-2 intensity start increasing from after completion of zippering (540 m.p.f.). Error bars represent standard error of the mean (S.E.M.). See *Figure 7—figure supplement 2* for individual RAB-3, CLA-1 and SYD-2 intensity values. Times are in m.p.f. (minutes post fertilization). (**K**) Schematic model showing progressive retrograde zippering leading to placement of the AIB neurite along two different layers. This is accompanied by a switch in SYG-1 expression between layers, and synaptic protein localization in a retrograde order along the neurite, resembling the order of zippering.

The online version of this article includes the following source data and figure supplement(s) for figure 7:

**Figure supplement 1.** Presynaptic proteins populate the distal neurite following zippering-mediated neurite placement.

**Figure supplement 2.** Individual data points from *Figure 7J*.

**Figure supplement 3.** Presynaptic protein distribution phenotype in the AIB neurite of *syg-1(ky652)*.

**Figure supplement 3—source data 1.** Mean fluorescence intensities of RAB-3 in adhered and detached regions of the AIB distal neurite in *syg-1(ky652)* (*Figure 7—figure supplement 3N*).

*supplement 1G-K*), indicating that molecules that affect synaptogenesis do not necessarily result in fasciculation defects for AIB. Also consistent with the importance of AIB neurite placement in the anterior neighborhood for correct synaptogenesis, we observed that in *syg-1(ky652)*, RAB-3 signal was specifically and consistently reduced in regions of the AIB distal neurite incapable of repositioning to the anterior neighborhood (*Figure 7—figure supplement 3*). Overall, our study identified a role for differential adhesion in regulating neurite placement via retrograde zippering, which in turn influences synaptic specificity onto target neurons (*Figure 7K*).

## SYG-1 is required for layer-specific placement of rich-club neuron AVE

We next examined if *syg-1* also mediates layer-specific placement of other neurites. We focused on the rich-club AVE neurons, the neurites of which are also placed in two neighborhoods, one of which coincides with the *syg-1*-enriched AIB distal neighborhood (*Figure 8A–D*) (*White et al., 1986*, *Towlson et al., 2013*, *Sabrin, 2019*, *Moyle et al., 2021*). Reconstructions from electron micrographs reveal that the AVE neurons have a morphology similar to AIB, however its neurite is more anteriorly placed (by one stratum) with respect to AIB (*Figure 8A*; *Moyle et al., 2021*). Therefore, the proximal neurite of AVE occupies the S2/S3 neighborhood (also occupied by the AIB distal neurite) (*Figure 8B–D*). Since *syg-1* expression is enriched in this 'AIB anterior/AVE posterior' neighborhood, we tested, by examining AVE neurite placement relative to the RIM neurons, if placement of the AVE neurite in this neighborhood is also affected in *syg-1(ky652)* mutants. When we fluorescently labeled RIM and AVE in wildtype animals, we observed that the proximal AVE neurite runs along the RIM neurite, consistent with EM studies (*White et al., 1986*; *Witvliet et al., 2021*, *Figure 8E, F and F'*). By contrast, in *syg-1* mutants the AVE proximal neurite frequently deviates from its trajectory along RIM (seen in 50 % of *syg-1(ky652)*) mutants versus 9.1 % in wild type (*Figure 8G, H, H'1*). The dorsal midline shift of AVE is also affected in *syg-1* mutant animals (mean length = 2.73 µm in *syg-1(ky652)* and 3.99 µm in wild-type animals; *Figure 8J*). The detachment of the AVE neurite resembles defects that would arise from defective zippering of the neurite onto this neighborhood. Together with the AIB studies, these observations are consistent with SYG-1 expression in a specific neuropil neighborhood resulting in specific sorting of neurites into the neighborhood by zippering mechanisms.

## Discussion

The precise assembly of the cellular architecture of AIB in the context of the layered nerve ring neuropil underwrites its role as a "rich-club" neuron. AIB was identified, through graph theory analyses, as a rich-club neuron (*Towlson et al., 2013*) - a connector hub with high betweeness centrality, which links nodes of the *C. elegans* neural networks with high efficiency. We observe that the AIB neurite segments are precisely placed on distinct functional layers of the nerve ring neuropil, and that the placement of these segments, in the context of the pre- and postsynaptic polarity of the neurite, enables AIB to receive inputs from one neighborhood and relay information to the other, thereby linking otherwise modular and functionally distinct layers. Our connectomic analyses and in vivo imaging reveal that these features of AIB architecture are stereotyped across examined *C. elegans* animals, even as early as the first larval stage, L1 (*Witvliet et al., 2021*). They are also evolutionarily conserved in nematodes, as examination of AIB in the connectome of the nematode *Pristionchus pacificus,* which is separated from *C. elegans* by 100 million years of evolutionary time, revealed similar design principles (*Hong et al., 2019*). The architecture of AIB is reminiscent to that seen for other 'nexus neurons' in layered neuropils, such as AII amacrine cells in the inner plexiform layer of the vertebrate retina (*Marc et al., 2014*). Like AIB, AII amacrine cells receive inputs from one laminar neighborhood (rod bipolar axon terminals in 'lower sublamina b') and produce outputs onto a different neighborhood (ganglion cell dendrites in 'sublamina a') (*Kolb, 1995*; *Strettoi et al., 1992*). For these nexus neurons, as for AIB, the precise placement within neuropil layers is critical for their function and connectivity. We now demonstrate that for AIB, this precise placement is governed via differential adhesion instructed by the layer-specific expression of IgCAM SYG-1. Interestingly, other 'rich-club' neurons that emerged from connectomic studies, such as AVE and RIB, are also placed along SYG-1-expressing nerve ring layers, suggesting that similar, SYG-1 dependent and layer-specific mechanisms could underpin placement of these neurons.

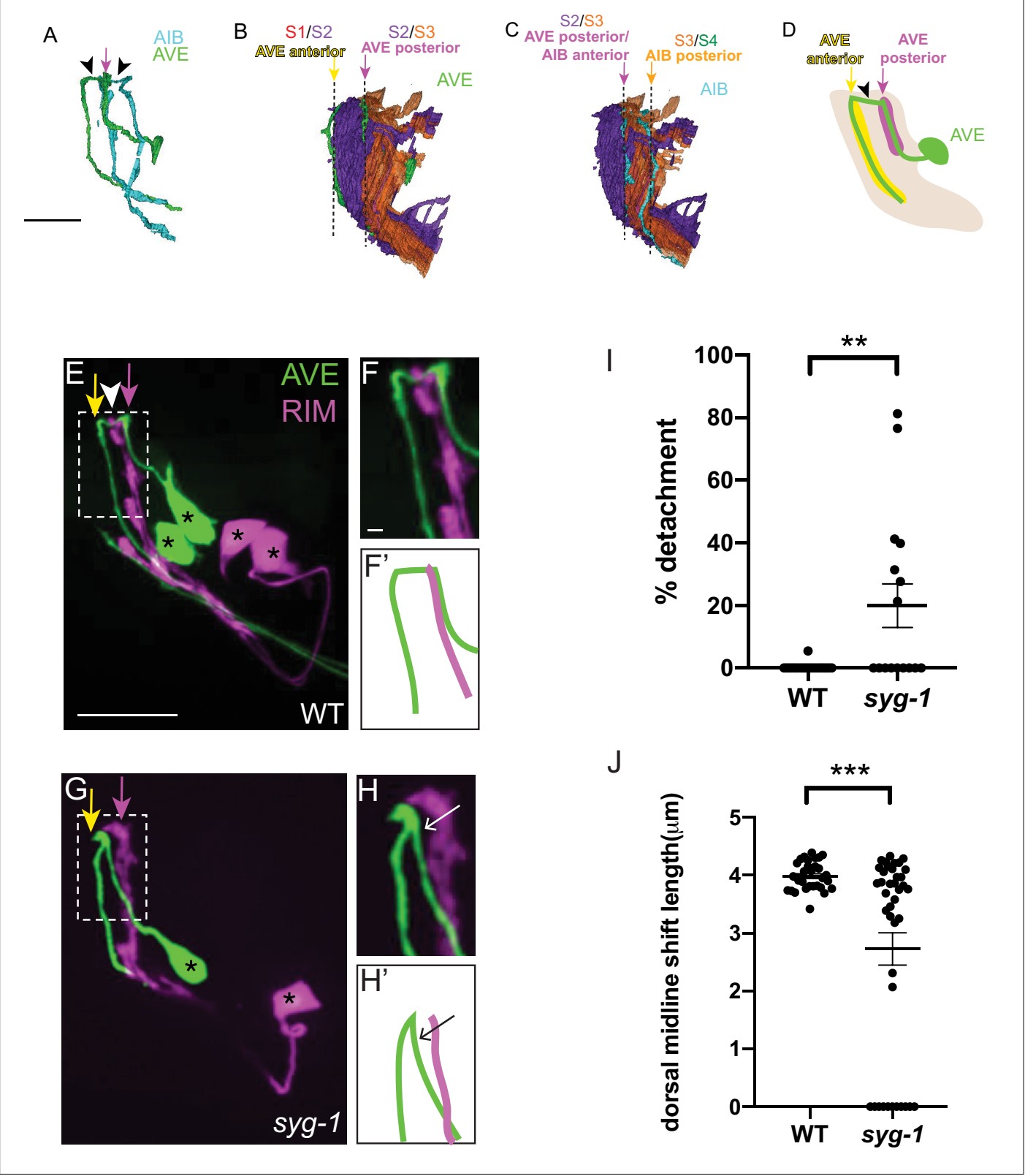

**Figure 8.** SYG-1 regulates neighborhood-specific placement of AVE. (**A**) Volumetric reconstruction of command interneuron AVER (green) and AIBR (cyan) from the segmented JSH EM dataset (*Brittin et al., 2018*; *White et al., 1986*). Note the similarity in morphology of the two neurons. The distal neurite of AIB and the proximal neurite of AVE lie at the same position (indicated by magenta arrow). The arrowheads indicate the dorsal shift that forms the chiasm in AIB and AVE. Scale bar = 5 µm, also applies to B. (**B,C**) Volumetric reconstruction of the AVE neurons (green) (**B**) and the AIB

*Figure 8 continued on next page*

Figure 8 continued

neurons (cyan) (**C**) in the context of the nerve ring strata S2 (purple) and S3 (orange). Note the placement of the AVE proximal neurite along the border of S2 and S3, and the AVE distal neurite at the anterior boundary of S2 (the anterior boundary abuts S1, not shown here). Note the placement of the AIB distal neurite, also at the S2/S3 border, similar to the AVE proximal neurite. The dashed lines indicate the layer borders. The yellow, magenta and orange arrows correspond to the S1/S2, S2/S3, and S3/S4 borders respectively (S4 not shown here). Scale bar in C = 5 μm. (**D**) Schematic of the lateral view of AVE (green) in the context of its neighborhoods: proximal (magenta) and distal (yellow), with the nerve ring (light brown) and pharynx (gray). Black arrowhead indicates a posterior-anterior chiasm. The magenta and yellow arrows indicate the positions of the AVE proximal and AVE distal neighborhoods, respectively and coincide with the S2/S3 and S1/S2 borders, respectively (see B). Note that while the design principles of AVE are similar to those of rich-club interneuron AIB, their positions in the nerve ring, and the strata they connect, are different – the AVE neurite is placed more anteriorly by one stratum compared to AIB. E,F,F', Confocal image of wild-type animal with AVE and RIM co-labeled. The magenta and yellow arrows indicate the positions of the AVE proximal and AVE distal neighborhoods, respectively. White arrowhead indicates AVE chiasm, corresponding to its anterior shift. Dashed box shows region of contact of the AVE and RIM neurites, magnified in F. (**F'**) is a schematic of the image in (**F**). Scale bar corresponds to 10 μm in E and 1 μm in F. Scale bars in E and F apply to G and H, respectively. Cell bodies are marked with an asterisk. G,H,H', As E,F,F' but in *syg-1(ky652)* mutant background. Note the gap between the AVE proximal neurite and the RIM neurites (**G,H'**) and defect in the dorsal midline shift. (**I**) Scatter plot showing quantification of the loss of contacts between the AVE and RIM neurites. The extent of detachment of the AVE proximal neurites from RIM, and hence its deviation from the RIM neighborhood, was quantified using the indicated formula in *Figure 4M* (also see STAR Methods). Scatter plot depicts % detachment values for wild type (n = 22) and *syg-1(ky652)* (n = 16) calculated from 11 and 8 animals respectively. Error bars indicate standard error of the mean (S.E.M.). **$P = 0.002$ (unpaired two-tailed t-test). Effect size d = 1.002. (**J**) Quantification of length of the posterior-anterior shift, quantified for each AVE neurite, for WT (n = 32) and *syg-1(ky652)* mutants (n = 40) and displayed as a scatter plot. These were calculated from 16 and 20 animals respectively from WT and *syg-1(ky652)*. Error bars indicate standard error of the mean (S.E.M.). ***$p = 0.0001$ (unpaired two-tailed t-test). Effect size d = 1.003. n represents the number of AIB neurites quantified.

Differential adhesion acts via retrograde zippering mechanisms to position AIB across multiple and specific layers. We established new imaging paradigms (*Wu et al., 2021*; *Wu et al., 2013*) to document in vivo embryonic development of AIB and observed that the sorting of its distal neurite segment onto the anterior neighborhood occurs, not via tip-directed fasciculation as we had anticipated, but via neurite-shaft retrograde zippering. Zippering mechanisms had been previously documented in tissue culture cells (*Barry et al., 2010*; *Voyiadjis et al., 2011*), where they were shown to act via biophysical forces of tension and adhesion (*Šmít et al., 2017*). However, these mechanisms have not been previously reported in vivo. We now demonstrate that retrograde zippering acts in vivo to precisely position neurites in specific neuropil layers. We observe that different segments of the AIB neurite are positioned in different neighborhoods by this mechanism. Zippering of the AIB neurite continues in the anterior neighborhood till adhesion of the neurite to this neighborhood exceeds the opposing action of mechanical tension on the neurite, and stops when adhesion and tension balance each other (*Figure 3*). Zippering stops at the dorsal midline where several neurites, including those of AIB's fasciculating partners, stop growing or change trajectories, possibly resulting in a change in adhesion forces on the AIB neurite, and a balance between adhesion and tension. Altogether, our data suggest that the interplay between biophysical forces results in precise placement of segments of the same neurite, allowing it to span two distinct neighborhoods.

Retrograde zippering depends on differential adhesion across layers and is instructed in part by the dynamic expression of SYG-1, and its interaction with the SYG-2 expressing AIB neurons. While we demonstrate that SYG-1 and SYG-2 are important for AIB neurite placement, we hypothesize that other adhesion molecules act redundantly in regulating placement, explaining the partial loss-of-function phenotypes observed in this study, and the gain of function phenotypes upon ectopic expression of SYG-1 in sublayers. Our work also demonstrates that differences in expression levels of IgCAMs such as SYG-1 can result in differential adhesion across whole neuropils. The observed role of SYG-1 in the nerve ring is reminiscent of the role of the SYG-1 and SYG-2 mammalian orthologs, Kirrel2 and Kirrel3, in axon sorting in the olfactory system (*Serizawa et al., 2006*), and consistent with observations in *C. elegans* that *syg-2* loss of function mutants result in defasciculation defects of the HSNL axon (*Shen et al., 2004*). Our findings are also consistent with studies on the roles of SYG-1 and SYG-2 *Drosophila* orthologues, Hibris and Roughest, in tissue morphogenesis of the pupal eye (*Bao and Cagan, 2005*). In these studies, Hibris and Roughest were shown to instruct complex morphogenic patterns by following simple, adhesion and surface energy-based biophysical principles that contributed to preferential adhesion of specific cell types. We now demonstrate that similar biophysical principles of differential adhesion might help organize neurite placement within heterogeneous tissues, such as neuropils in nervous systems.

SYG-1 and SYG-2 coordinate developmental processes that result in synaptic specificity for the AIB interneurons. Synapses in *C. elegans* are formed *en passant*, or along the length of the axon, similar to how they are assembled in the CNS for many circuits (*Jontes et al., 2000*; *Koestinger et al., 2017*). Placement of neurites within layers therefore restrict synaptic partner choice. We examined how these events of placement, and synaptogenesis, were coordinated for the AIB interneurons and observed coincidence of presynaptic assembly and retrograde zippering of the AIB neurite. SYG-1 and SYG-2 were identified in *C. elegans* for their role in synaptic specificity (*Shen and Bargmann, 2003*; *Shen et al., 2004*), and the assembly of synaptic specializations can result in changes in the cytoskeletal structure and adhesion junctions (*Missler et al., 2012*). We observe in our studies that zippering precedes the (detectable) subcellular localization of presynaptic components, suggesting that during AIB development, neurite placement by retrograde zippering constitutes a specificity step distinct from synaptic protein localization and synapse formation. Nonetheless, we hypothesize that coordinated assembly of synaptic sites during the process of retrograde zippering could provide forces that stabilize zippered stretches of the neurite. These could in turn 'button' and fasten the AIB neurite onto the anterior layer, securing its relationship with its postsynaptic partner. Consistent with this hypothesis, we observe that ablation of one of its main postsynaptic partners, the RIM neurons, results in defects in AIB placement in the anterior neighborhood. Given the important role of adhesion molecules in coordinating cell-cell interactions and synaptogenesis (*Sanes and Zipursky, 2010*; *Sanes and Zipursky, 2020*; *Tan et al., 2015*; *Yamagata and Sanes, 2008*; *Yamagata and Sanes, 2012*), we speculate that adhesion molecules involved in synaptogenesis and neurite placement within layered neuropils might similarly act to coordinate differential adhesion and synaptogenesis onto target neurons.

Zippering mechanisms via affinity-mediated adhesion might help instruct neighborhood coherence while preserving 'fluid', or transient interactions among neurites within neuropil structures. Analysis of connectome data and examination of neuronal adjacencies within the nerve ring neuropil revealed that contact profiles for single neurons vary across animals, indicative of fluid or transient interactions during development (*Moyle et al., 2021*). Yet neuropils have a stereotyped and layered architecture encompassing specific circuits. We hypothesize that dynamic expression of adhesion molecules help preserve tissue organization in tangled neuropils via the creation of affinity relationships of relative strengths. These relationships, in the context of outgrowth decisions of single neurites, would contribute to the sorting of neurites onto specific strata. We propose that sorting of neurite into strata would happen through biophysical interactions not unlike those reported for morphogenic events in early embryos and occurring via differential adhesion (*Steinberg, 1962*; *Steinberg and Gilbert, 2004*). Spatiotemporally restricted expression of CAMs in layers, as we observe for SYG-1 and has been observed for other CAMs in layered neuropils (*Sanes and Zipursky, 2010*; *Sanes and Zipursky, 2020*; *Tan et al., 2015*; *Yamagata and Sanes, 2008*; *Yamagata and Sanes, 2012*) would then result in dynamic, affinity-mediated relationships that preserve neighborhood coherence in the context of 'fluid', or transient interactions among neurites within the neuropil structures.

## Materials and methods

### Key resources table

| Reagent type (species) or resource | Designation | Source or reference | Identifiers | Additional information |
|---|---|---|---|---|
| Strain (*C. elegans*) | ujIs113[pie-1p::mCherry::H2B::pie-1 3'UTR+ nhr-2p::his-24::mCherry::let-858 3'UTR+ unc-119(+)];II | *Duncan et al., 2019* | BV276 | Strain available from D. Colón-Ramos lab |
| Strain (*C. elegans*) | ujIs113;oyIs48[Pceh-36::GFP, lin-15(+)];V | gift from John Murray | JIM158 | Strain available from D. Colón-Ramos lab |
| Strain (*C. elegans*) | olaIs67[DACR2245 at 40 ng/uL + DACR1412 at 30 ng/uL + DACR218 at 30 ng/uL];X | This paper | DCR5516 | Strain available from D. Colón-Ramos lab |
| Strain (*C. elegans*) | olaex3394[DACR2796 at 60 ng/uL + DACR2651 at 60 ng/uL + DACR218 at 30 ng/uL] | This paper | DCR5761 | Strain available from D. Colón-Ramos lab |
| Strain (*C. elegans*) | olaex3666[DACR199 at 2 ng/uL + DACR218 at 30 ng/uL];olaIs67 | This paper | DCR6222 | Strain available from D. Colón-Ramos lab |

*Continued on next page*

*Continued*

| Reagent type (species) or resource | Designation | Source or reference | Identifiers | Additional information |
|---|---|---|---|---|
| Strain (*C. elegans*) | oyIs48;olaIs67 | This paper | DCR6301 | Strain available from D. Colón-Ramos lab |
| Strain (*C. elegans*) | olaex4624[DACR3149 at 10 ng/uL + DACR218 at 30 ng/uL];olaIs67 | This paper | DCR7648 | Strain available from D. Colón-Ramos lab |
| Strain (*C. elegans*) | olaIs68[DACR2245 at 40 ng/uL + DACR1412 at 30 ng/uL + DACR218 at 30 ng/uL] | This paper | DCR5517 | Strain available from D. Colón-Ramos lab |
| Strain (*C. elegans*) | olaIs68;*syg-1(ky652)* | This paper | DCR8220 | Strain available from D. Colón-Ramos lab |
| Strain (*C. elegans*) | olaIs68;*syg-1(ok3640)* | This paper | DCR8486 | Strain available from D. Colón-Ramos lab |
| Strain (*C. elegans*) | olaex4624;olaIs68;*syg-1(ky652)* | This paper | DCR8183 | Strain available from D. Colón-Ramos lab |
| Strain (*C. elegans*) | kyIs235;kyEx679;*syg-1(ky652)* | This paper | CX5862 | Strain available from D. Colón-Ramos lab |
| Strain (*C. elegans*) | kyEx679;olaIs68;*syg-1(ky652)* | This paper | DCR8180 | Strain available from D. Colón-Ramos lab |
| Strain (*C. elegans*) | olaex4624;kyEx679;olaIs68;*syg-1(ky652)* | This paper | DCR8489 | Strain available from D. Colón-Ramos lab |
| Strain, strain background (*C. elegans*) | olaIs68;*syg-2(ky671)* | This paper | DCR6767 | Strain available from D. Colón-Ramos lab |
| Strain (*C. elegans*) | olaex4624;olaIs68;*syg-2(ky671)* | This paper | DCR8468 | Strain available from D. Colón-Ramos lab |
| Strain (*C. elegans*) | oyIs48; olaIs68;*syg-1(ky652)* | This paper | DCR8488 | Strain available from D. Colón-Ramos lab |
| Strain (*C. elegans*) | olaex5120[DACR3529 at 30 ng/uL + DACR1412 at 30 ng/uL + DACR218 at 30 ng/uL] | This paper | DCR8440 | Strain available from D. Colón-Ramos lab |
| Strain (*C. elegans*) | olaex5063[DACR3492 at 25 ng/uL + DACR3505 at 40 ng/uL + DACR2312 at 25 ng/uL + DACR20 at 25 ng/uL];olaIs67 | This paper | DCR8365 | Strain available from D. Colón-Ramos lab |
| Strain (*C. elegans*) | olaex4071[DACR2637 at 15 ng/uL + DACR218 at 30 ng/uL];olaIs67 | This paper | DCR6814 | Strain available from D. Colón-Ramos lab |
| Strain (*C. elegans*) | olaex4130[DACR2704 at 100 ng/uL + DACR218 at 50 ng/uL];ujIs113 | This paper | DCR6920 | Strain available from D. Colón-Ramos lab |
| Strain (*C. elegans*) | olaex4052[DACR2607 at 100 ng/uL + DACR2609 at 25 ng/uL + DACR218 at 30 ng/uL];olaIs67 | This paper | DCR6782 | Strain available from D. Colón-Ramos lab |
| Strain (*C. elegans*) | olaex4054[DACR2607 at 100 ng/uL + DACR2609 at 25 ng/uL + DACR218 at 30 ng/uL];olaIs67 | This paper | DCR6784 | Strain available from D. Colón-Ramos lab |
| Strain (*C. elegans*) | olaex3388[DACR2371 at 75 ng/uL + DACR2404 at 30 ng/uL + DACR218 at 30 ng/uL] | This paper | DCR5730 | Strain available from D. Colón-Ramos lab |
| Strain (*C. elegans*) | olaex4618[DACR2607 at 100 ng/uL + DACR2609 at 25 ng/uL + DACR2863 at 25 ng/uL + DACR218 at 30 ng/uL];olaIs67 | This paper | DCR7642 | Strain available from D. Colón-Ramos lab |
| Strain (*C. elegans*) | olaex4619[DACR2607 at 100 ng/uL + DACR2609 at 25 ng/uL + DACR2863 at 25 ng/uL + DACR218 at 30 ng/uL];olaIs67 | This paper | DCR7643 | Strain available from D. Colón-Ramos lab |
| Strain (*C. elegans*) | olaex3949[DACR2607 at 100 ng/uL + DACR2609 at 25 ng/uL + DACR2351 at 25 ng/uL + DACR218 at 30 ng/uL] | *Moyle et al., 2021* | DCR6633 | Strain available from D. Colón-Ramos lab |
| Strain (*C. elegans*) | olaex2887[DACR2245 at 100 ng/uL + DACR2404 at 30 ng/uL + DACR218 at 30 ng/uL] | This paper | DCR4894 | Strain available from D. Colón-Ramos lab |
| Strain (*C. elegans*) | olaex3570[DACR2481 at 10 ng/uL + DACR218 at 50 ng/uL];ujIs113 | This paper | DCR6082 | Strain available from D. Colón-Ramos lab |

*Continued on next page*

*Continued*

| Reagent type (species) or resource | Designation | Source or reference | Identifiers | Additional information |
| --- | --- | --- | --- | --- |
| Strain (*C. elegans*) | olaex5105[DACR3605 at 50 ng/uL + DACR218 at 30 ng/uL] | This paper | DCR8421 | Strain available from D. Colón-Ramos lab |
| Strain (*C. elegans*) | olaIs117[DACR3502 at 30 ng/uL + DACR20 at 25 ng/uL];olaIs68 | This paper | DCR8347 | Strain available from D. Colón-Ramos lab |
| Strain (*C. elegans*) | olaIs117;olaIs68;*syg-1(ky652)* | This paper | DCR8350 | Strain available from D. Colón-Ramos lab |
| Strain (*C. elegans*) | olaex5059[DACR3503 at 10 ng/uL + DACR20 at 25 ng/uL];olaIs68;*syg-1(ky652)* | This paper | DCR8361 | Strain available from D. Colón-Ramos lab |
| Strain (*C. elegans*) | olaex5050[DACR3698 at 30 ng/uL + DACR20 at 25 ng/uL];olaIs68;*syg-1(ky652)* | This paper | DCR8352 | Strain available from D. Colón-Ramos lab |
| Strain (*C. elegans*) | oyIs48; olaex5059; olaIs68; *syg-1(ky652)* | This paper | DCR8470 | Strain available from D. Colón-Ramos lab |
| Strain (*C. elegans*) | oyIs48; olaIs117; olaIs68; *syg-1(ky652)* | This paper | DCR8472 | Strain available from D. Colón-Ramos lab |
| Strain (*C. elegans*) | olaex4087[DACR1412 at 30 ng/uL + DACR2618 at 50 ng/uL + DACR218 at 30 ng/uL] | This paper | DCR6841 | Strain available from D. Colón-Ramos lab |
| Strain (*C. elegans*) | oyIs48;olaIs68;*syg-2(ky671)*; | This paper | DCR8758 | Strain available from D. Colón-Ramos lab |
| Strain (*C. elegans*) | olaEx5279[DACR3527 at 30 ng/uL + DACR20 at 25 ng/uL]; olaIs68; *syg-1(ky652)* | This paper | DCR8762 | Strain available from D. Colón-Ramos lab |
| Strain (*C. elegans*) | olaEx5276 [DACR3780 at 5 ng/ul + DACR1412 at 20 ng/uL + DACR218 at 30 ng/uL] | This paper | DCR8759 | Strain available from D. Colón-Ramos lab |
| Strain (*C. elegans*) | olaEx5281[DACR3888 at 30 ng/uL + DACR20 at 30 ng/uL]; olaIs68; *syg-2(ky671)* | This paper | DCR8764 | Strain available from D. Colón-Ramos lab |
| Strain (*C. elegans*) | olaEx5283[DACR3781 at 30 ng/uL + DACR20 at 25 ng/uL]; olaIs68;*syg-1(ky652)* | This paper | DCR8766 | Strain available from D. Colón-Ramos lab |
| Strain (*C. elegans*) | olaIs117; olaIs68; *syg-1(ky652); syg-2(ky671)* | This paper | DCR8767 | Strain available from D. Colón-Ramos lab |
| Strain (*C. elegans*) | zbIs3[cnd-1p::PH::GFP] | *Fan et al., 2019* | BV293 | Strain available from D. Colón-Ramos lab |
| Strain (*C. elegans*) | zbIs3;olaIs68;*syg-1(ky652)* | This paper | DCR8772 | Strain available from D. Colón-Ramos lab |
| Strain (*C. elegans*) | kyex684[syg-2:GFP] | *Shen et al., 2004* | TV6006 | Strain available from D. Colón-Ramos lab |
| Strain (*C. elegans*) | olaex5347[DACR1412 at 30 ng/uL + DACR218 at 30 ng/uL] | This paper | DCR8922 | Strain available from D. Colón-Ramos lab |
| Strain (*C. elegans*) | olaex5332[DACR3901 at 125 ng/uL + DACR2404 at 75 ng/uL + DACR218 at 30 ng/uL] | This paper | DCR8894 | Strain available from D. Colón-Ramos lab |
| Strain (*C. elegans*) | olaex5340[DACR3890 at 100 ng/uL + DACR2404 at 75 ng/uL + DACR218 at 30 ng/uL] | This paper | DCR8908 | Strain available from D. Colón-Ramos lab |
| Strain (*C. elegans*) | olaex5144[DACR3492 at 50 ng/uL + DACR3493 at 50 ng/uL + DACR218 at 30 ng/uL];olaIs67 | This paper | DCR8469 | Strain available from D. Colón-Ramos lab |
| Strain (*C. elegans*) | olaex5195[DACR3529 at 30 ng/uL + DACR218 at 30 ng/uL];ujIs113 | This paper | DCR8626 | Strain available from D. Colón-Ramos lab |
| Strain (*C. elegans*) | olaIs67;*syd-2(ola341)* | This paper | DCR6756 | Strain available from D. Colón-Ramos lab |
| Strain (*C. elegans*) | olaex3666; olaIs67;*syd-2(ola341)* | This paper | DCR6842 | Strain available from D. Colón-Ramos lab |

*Continued on next page*

*Continued*

| Reagent type (species) or resource | Designation | Source or reference | Identifiers | Additional information |
|---|---|---|---|---|
| Strain (*C. elegans*) | hdIs32 [glr-1::DsRed2]. gvEx173 [opt-3::GFP+ rol-6(su1006)] | CGC | NC1750 | Strain available from D. Colón-Ramos lab |
| Strain (*C. elegans*) | gvex173;*syg-1(ky652)* | This paper | DCR8907 | Strain available from D. Colón-Ramos lab |

## Materials availability

See *Supplementary file 1* for plasmids generated and used in this study. See Key Resources Table for *C. elegans* strains used in this study.

## Code availability

From previously determined adjacencies (*Brittin et al., 2018*; *Brittin et al., 2021*; *Witvliet et al., 2021*), cosine similarities were calculated in Excel, using the formula described in Materials and methods. For computing binary connection matrices for centrality analysis (detailed in Materials and methods below). we used the function "betweenness_bin.m" in the Brain Connectivity Toolbox (*Rubinov and Sporns, 2010*) of MATLAB2020.

## Maintenance of *C. elegans* strains

*C. elegans* strains were raised at 20 °C using OP50 *Escherichia coli* seeded on NGM plates. N2 Bristol is the wild-type reference strain used.

## Molecular biology and generation of transgenic lines

We used Gibson Assembly (New England Biolabs) or the Gateway system (Invitrogen) to make plasmids (*Supplementary file 1*) used for generating transgenic *C. elegans* strains (Key Resources Table). Detailed cloning information or plasmid maps will be provided upon request. Transgenic strains were generated via microinjection with the construct of interest at 2–100 ng/μL by standard techniques (*Mello and Fire, 1995*). Co-injection markers *unc-122p*: GFP or *unc-122p*: RFP were used.

We generated the *syg-1* transcriptional reporter (*Figure 5*, *Figure 5—figure supplement 1*) by fusing membrane-targeted PH:GFP to a 3.5 kb *syg-1* promoter region as described (*Schwarz et al., 2009*). The translational reporter was generated by fusing a GFP-tagged *syg-1b* cDNA using the same promoter (*Figure 5*). For cell-specific SYG-1 expression, full-length SYG-1, SYG-1 ecto (extracellular+ TM domain - amino acids 1–574, *Chao and Shen, 2008*) or SYG-1 endo (signal peptide+ TM domain+ cytoplasmic domain – amino acids 1–31 + 526-574) were used.

For cell-specific labeling and expression in larvae, we used an *inx-1* promoter for AIB (*Altun and Chen, 2008*), a *ceh-36* promoter for AWC and ASE (*Kim et al., 2010*), *tdc-1*, *gcy-13* and *cex-1* promoters for RIM (*Greer et al., 2008*; *Piggott et al., 2011*), and an *opt-3* promoter for AVE (https://www.wormatlas.org).

## SNP mapping and whole-genome sequencing

We performed a visual forward genetic screen in an integrated wild type transgenic strain (*olaIs67*) with AIB labeled with cytoplasmic mCherry and AIB presynaptic sites labeled with GFP:RAB-3. Ethyl methanesulfonate (EMS) mutagenesis was performed and animals were screened for defects in placement of the AIB neurite, or presynaptic distribution. We screened for these same phenotypes in our reverse genetic screens as well, where we crossed the marker strain (*olaIs67*) to characterized mutant alleles. We screened F2 progeny on a Leica DM 5000 B compound microscope with an HCX PL APO 63 x/1.40–0.60 oil objective.

Mutants from forward genetic screens *were* out-crossed six times to wild type (N2) animals and mapped via single-nucleotide polymorphism (SNP) (*Davis et al., 2005*) and whole-genome sequencing as previously described (*Sarin et al., 2008*). We analyzed the results using the Galaxy platform (https://galaxyproject.org/news/cloud-map/, EMS variant density mapping workflow *Minevich et al., 2012*). Our forward genetic screens uncovered 19 mutants with neurite placement defects and 12 with synaptic defects, including *syd-2(ola341)* (*Figure 7—figure supplement 1*).

We also performed a reverse genetic screen with candidate adhesion molecules expressed in AIB and its primary postsynaptic partner, RIM (*Schwarz et al., 2009*), for defects in AIB neurite placement and presynaptic pattern. Of these mutants *syg-1(ky652)* and *syg-2(ky671)* exhibited AIB neurite placement defects.

## Confocal imaging of *C. elegans* larvae and image processing

We used an UltraView VoX spinning disc confocal microscope with a 60 x CFI Plan Apo VC, NA 1.4, oil objective on a NikonTi-E stand (PerkinElmer) with a Hamamatsu C9100–50 camera. We imaged the following fluorescently tagged fusion proteins, eGFP, GFP, PH:GFP (membrane-tethered), RFP, mTagBFP1, mCherry, mCherry:PH, mScarlet, mScarlet:PH at 405, 488 or 561 nm excitation wavelength. We anesthetized larval stage four animals (unless otherwise mentioned) at room temperature in 10 mM levamisole (Sigma) and mounted them on glass slides for imaging. For *Figure 5* and the RIM neuron ablation images in *Figure 5—figure supplement 4*, larval stage three animals were imaged.

We used the Volocity image acquisition software (Improvision by Perkin Elmer) and processed our images using Fiji (*Schindelin et al., 2012*). Image processing included maximum intensity projection, 3D projection, rotation, cropping, brightness/contrast, line segment straightening, and pseudo coloring. All quantifications from confocal images were conducted on maximal projections of the raw data. Pseudocoloring of AIBL and AIBR was performed in Fiji. To achieve this, pixels corresponding to the neurite of either AIBL/R were identified and the rest of the pixels in the image were cleared. This was done for both neurons of the pair and the resulting images were merged. For quantifications from confocal images, n = number of neurons quantified, unless otherwise mentioned.

## Embryo labeling, imaging, and image processing

For labeling of neurites in embryos, we used membrane tethered PH:GFP or mScarlet:PH. A subtractive labeling strategy was employed for AIB embryo labeling (*Figure 2—figure supplement 2A-C*; *Armenti et al., 2014*; *Moyle et al., 2021*). Briefly, we generated a strain containing unc-42p::ZF1::PH::GFP and lim-4p::SL2::ZIF-1, which degraded GFP in the sublateral neurons, leaving GFP expression only in the AIB and/or ASH neurons. Onset of twitching was used as a reference to time developmental events. Embryonic twitching is stereotyped and starts at 430 min post fertilization (m.p.f) for our imaging conditions.

Embryonic imaging was performed via dual-view inverted light sheet microscopy (diSPIM) (*Kumar et al., 2014*; *Wu et al., 2013*) and a combined triple-view line scanning confocal/DL for denoising (*Wu et al., 2021*, also described below) described below. Images were processed and quantifications from images were done using CytoSHOW, an open-source image analysis software. CytoSHOW can be downloaded from http://www.cytoshow.org/ as described (*Duncan et al., 2019*).

## Triple-view line-scanning confocal/DL

We developed a triple-view microscope that can sequentially capture three specimen views, each acquired using line-scanning confocal microscopy (*Wu et al., 2021*). Multiview registration and deconvolution can be used to fuse the three views (*Wu et al., 2016*), improving spatial resolution. Much of the hardware for this system is similar to the previously published triple-view system (*Wu et al., 2016*), that is we used two 0.8 NA water immersion objectives for the top views and a 1.2 NA water immersion lens placed beneath the coverslip for the bottom view. To increase acquisition speed and reduce photobleaching, we applied a deep-learning framework (*Weigert et al., 2018*) to predict the triple-view result when only using data acquired from the bottom view. The training datasets were established from 50 embryos (anesthetized with 0.3 % sodium azide) in the post-twitching stage, in which the ground truth data were the deconvolved triple view confocal images, and the input data were the raw single view confocal images. These approaches resulted in improved resolution (270nm X 250 nm X 335 nm).

## Cell lineaging

Cell lineaging was performed using StarryNite/AceTree (*Bao et al., 2006*; *Boyle et al., 2006*; *Murray et al., 2006*). Light sheet microscopy and lineaging approaches were integrated to uncover cell identities in pre-twitching embryos (*Duncan et al., 2019*). Lineaging information for promoters is available at http://promoters.wormguides.org. Our integrated imaging and lineaging approaches enabled us

to identify a promoter region of *inx-19* which is expressed in the RIM neurons prior to RIM neurite outgrowth (~370 m.p.f.) and in additional neurons in later embryonic stages. The *inx-19p* was one of the promoters used for embryonic ablation of the RIM neurons (described in the next section).

Our integrated imaging and lineaging approach also enabled us to identify two promoters with expression primarily in neurons located at the AIB posterior neighborhood (*nphp-4*p and *mgl-1b*p). 4/4 neuron classes that were identified to have *nphp4*p expression are in the AIB posterior neighborhood (ADL/R, ASGL/R, ASHL/R, ASJL/R) and 2/3 neuron classes that were identified to have *mgl-1b*p expression are in the AIB posterior neighborhood (AIAL/R, ADFR) (http://promoters.wormguides.org). We used these promoters to drive ectopic expression of a *syg-1* cDNA specifically in the posterior neighborhood.

We also used this imaging and lineaging approach to identify SYG-1 expressing neurons in the anterior and posterior neighborhoods (*Figure 5—figure supplement 2*). We determined cell identities by lineaging both sides of an embryo expressing the *syg-1* transcriptional reporter (see the 'Molecular Biology and generation of transgenic lines' section). The cell identities obtained for the left and right sides of the nerve ring were consistent.

## Caspase-mediated ablation of RIM neurons

The RIM neurons were ablated using a split-caspase ablation system (*Chelur and Chalfie, 2007*). We generated one set of transgenic strains with co-expression of the p12 and p17 subunit of human Caspase-3, both expressed under *inx-19p* (termed ablation strategy 1), and another set of ablation strains with co-expression of the p12 subunit expressed under *inx-19p* and p17 under *tdc-1p* (termed ablation strategy 2) (*Figure 5—figure supplement 4*). L3 larvae from the RIM-ablated populations were imaged on the spinning-disk confocal microscope (described in the **'Confocal imaging of *C. elegans* larvae and image processing'** section).

## Rendering of neurites and contacts in the EM datasets

From available EM datasets (*Brittin et al., 2021*; *Cook et al., 2019*; *White et al., 1986*; *Witvliet et al., 2021*) we rendered the segmentations of neuron boundaries in 2D using TrakEM2 in Fiji. TrakEM2 segmentations were volumetrically rendered by using the 3D viewer plugin in Fiji (downloaded from https://imagej.net/Fiji#Downloads) and saved as object files (.obj), or by using the 3D viewer in CytoSHOW.

To generate 3D mappings of inter-neurite membrane contact, the entire collection of 76,046 segmented neuron membrane boundaries from the JSH TEM datasets (*Brittin et al., 2018*; *White et al., 1986*) were imported from TrakEM2 format into CytoSHOW as 2D cell-name-labelled and uniquely color-coded regions of interest (ROIs). To test for membrane juxtaposition, we dilated each individual cell-specific ROI by nine pixels (40.5 nm) and identified for overlap by comparing with neighboring undilated ROIs from the same EM slice. A collection of 289,012 regions of overlap were recorded as new ROIs, each bearing the color code of the dilated ROI and labeled with both cell-names from the pair of the overlapped ROIs. These 'contact patch' ROIs were then grouped by cell-pair-name and rendered via a marching cubes algorithm to yield 3D isosurfaces saved in.obj files. Each of the 8852 rendered.obj files represents all patches of close adjacency between a given pair of neurons, color-coded and labeled by cell-pair name. Selected.obj files were co-displayed in a CytoSHOW3D viewer window to produce views presented in *Figure 1*, *Figure 1—figure supplement 1* and *Figure 1—figure supplement 2*.

## Schematic representation of larval *C. elegans*

The schematic representations of larval *C. elegans* in *Figure 1* and *Figure 5—figure supplement 3* were made using the 3D worm model in OpenWorm (http://openworm.org - 3D Model by Christian Grove, WormBase, CalTech).

## Quantification and statistical analysis

## Cosine similarity analysis for comparing AIB contacts across connectomes

We performed cosine similarity analysis (*Han et al., 2012*) on AIB contacts in available connectome datasets (*Brittin et al., 2021*; *White et al., 1986*; *Witvliet et al., 2021*). For each available adjacency dataset (*Brittin et al., 2021*; *Moyle et al., 2021*; *Witvliet et al., 2021*), we extracted

vectors comprising of the weights of AIB contacts with neurons common to all the datasets. We then performed cosine similarity analysis on these vectors using the formula:

$$\frac{\sum_{i=1}^{n} A_i B_i}{\sqrt{\sum_{i=1}^{n} A_i^2} \sqrt{\sum_{i=1}^{n} B_i^2}}$$

where A and B are the two vectors under consideration with the symbol " denoting the i-th entry of each vector. The similarity values were plotted as a heat map for AIBL and AIBR using Prism. For the datasets L1_0 hr, L1_5 hr, L1_8 hr, L2_23 hr, L3_27 hr, L4_JSH and Adult_N2U, only the neuron-neuron contacts in the EM sections corresponding to the nerve ring were used (as opposed to the whole connectome).

### Betweenness centrality analysis

We analyzed betweenness centrality for two of the available connectomes of different developmental stages (L1 and adult) (*Witvliet et al., 2021*). By treating individual components (neurons) of a connectome as the vertices of a graph, we use the following definition of Betweenness Centrality for a vertex $v$,

$$v_{zip} + v_{unzip} = \frac{S_{anterior} - S_{posterior}}{\eta} - \frac{T_{anterior} - T_{posterior}}{\eta}(1 - cos\theta)$$

Here $\lambda_{st}(v)$ denotes the number of shortest paths between the vertices $s$ and $t$, that include vertex $v$, whereas $\lambda_{st}$ denotes the total number of shortest paths between the vertices $s$ and $t$. We finally divide $BC(v)$ by $(N-1)(N-2)/2$ to normalize it to lie between 0 and 1. For our implementation we use the Brain Connectivity Toolbox (*Rubinov and Sporns, 2010*) of MATLAB2020, in particular, the function "betweenness_bin.m" in which we input the binary connectivity matrix (threshold = 0) (*Fornito et al., 2016*) corresponding to the L1 and adult connectomes (*Witvliet et al., 2021*). We made a Prism box plot (10–90 percentile) of betweenness centrality values of all neurons in each of the two connectomes and highlighted the betweenness centrality values for AIBL and AIBR.

### Representation of AIB from confocal images

Since we observed that the proximal and distal neurites of AIBL and AIBR completely align and overlap (*Figure 1—figure supplement 1*) in confocal image stacks where the worms are oriented on their side, for representation purposes we have used the upper 50 % of z-slices in confocal image stacks to make maximum intensity projections. This shows the proximal neurite of AIBL in the context of the distal of AIBR (which has the same anterior-posterior position as the distal neurite of AIBL) (*Figure 1—figure supplement 1*), or vice versa. We used the same procedure for AVEL and AVER.

## Quantification of penetrance of AIB neurite placement defects and gain-of-function phenotypes

The penetrance of defects in AIB neurite placement in the anterior neighborhood in mutant (or ablation) strains was determined by visualizing the AIB neurite and scoring animals with normal or defective anterior neighborhood placement under the Leica compound microscope described. Animals in which the entire distal neurite was placed at a uniform distance from the proximal neurite, for both AIBL and AIBR, were scored as having normal AIB distal neurite placement.

The penetrance of the gain-of-function effects in ectopic SYG-1 expression strains was determined by scoring the percentage of animals showing ectopic AIB distal neurite placement in the posterior neighborhood. Animals with part (or whole) of the AIB distal neurite overlapping with the posterior neighborhood were considered as having ectopic AIB placement.

## Quantification of minimum perpendicular distance between neurites

Minimum perpendicular distances between neurites (*Figure 4—figure supplement 1F*, *Figure 5—figure supplement 4O*) were measured by creating a straight line selection (on Fiji) between the neurites (perpendicular to one of the neurites) in the region where the gap between them is estimated to be the smallest. The measurements were done on maximum intensity projections of raw confocal image stacks where the worms are oriented on their side (z-stacks acquired along left-right axis of the worm, producing a lateral view of the neurons).

## Quantification of percent detachment between neurites

The percent detachment for defasciculated neurites (AIB or AVE and RIM) is calculated by the formula % detachment = detached length ($L_d$) x 100/ total length ($L_t$) (also shown in *Figure 4M*). $L_d$ is calculated by making a freehand line selection along the detached region of the RIM neurite and measuring its length and $L_t$ is calculated by making a freehand selection along the RIM neurite for the entire length over which it contacts AIB or AVE, and measuring the length of the selection. All the measurements were performed on maximum intensity projections of confocal image stacks where the worms are oriented on their side (z-stacks acquired along left right axis of the worm, producing a lateral view of the neurons).

## Quantification of percentage of distal neurite placed in posterior neighborhood

Freehand line selections of the entire distal neurite ($L_t$) and only the portion of the distal neurite positioned in the posterior neighborhood ($L_p$) are measured using Fiji. ($L_p/L_t$)x100 provides the percentage of the distal neurite placed in the posterior neighborhood.

## Quantification of relative enrichment of SYG-1 reporter expression in the anterior neighborhood

Relative (anterior) enrichment of *syg-1* reporter expression in embryos (*Figure 5S*) is calculated using the formula, relative enrichment (*syg-1p*) = mean anterior neighborhood intensity ($I_a$)/mean posterior neighborhood intensity ($I_p$). These measurements were done in transgenic embryos co-expressing the AIB reporter and the *syg-1* transcriptional reporter. For calculation of $I_p$, a freehand line selection was made (using CytoSHOW, http://www.cytoshow.org/, *Duncan et al., 2019*) along the posterior band of *syg-1* expression and mean intensity along the selection was calculated. Same was done for calculation of $I_a$. The ratios of $I_a$ and $I_p$ were plotted as relative (anterior) enrichment values (*Figure 5S*). These values were calculated from 3D projections of deconvolved diSPIM images acquired at intensities within dynamic range (not saturated) at timepoints during embryogenesis (485, 515, and 535 min post fertilization), when the AIB neurite grows and is placed into the posterior and anterior neighborhoods. $I_a/I_p$ was calculated from the anterior and posterior *syg-1* bands on each side of the embryonic nerve ring per embryo (number of embryos = 4, number of $I_a/I_p$ values = 8).

## Quantification of the dorsal midline shift (chiasm) length of AIB

The dorsal midline shift (chiasm) lengths of AIB and AVE were calculated by making 3D maximum intensity projections of confocal z-stacks and orienting the neuron pair to a dorsal-ventral view. A straight line selection is made along the posterior-anterior shift of each neuron, and each arm of the 'X' of the chiasm was measured (using Fiji).

## Quantification of distal neurite length of AIB

The length of the distal neurite of AIB was measured by drawing a freehand line along the neurite segment occupying the distal neighborhood (including the chiasm) in maximum intensity projections of confocal image stacks where the worms are oriented on their side (z-stacks acquired along left-right axis of the worm, producing a lateral view of the neurons).

## Quantification of positions and velocities of the AIB neurite during embryogenesis

The positions of the AIB neurite in the anterior and posterior neighborhoods in *Figure 3C* are calculated from deconvolved maximum intensity projections of diSPIM images where the neurons are oriented in an axial view. These positions are determined by measuring the lengths along the AIB neurite from the unzippering/zippering forks to the dorsal midline. The distance of the zippering fork from the midline is subtracted from the total length of the neurite at the start of zippering, to obtain the length of the AIB neurite that has already zippered. The fraction of the length of the AIB neurite that has zippered to the initial length of the relocating AIB distal neurite, multiplied by 100, yields the percentage of the AIB neurite that has zippered at each timepoint. The reported values (in *Figure 5S*) of the percentages of the AIB neurite that has zippered are averages across the three independent

embryo datasets (used for the *Figure 3* plots). Embryos in which the AIB and RIM neurons were specifically labeled by the subtractive labeling strategy were used for the analysis. Reported measurements represent AIB neurites which were visible through the imaging window. Zippering velocity (*Figure 3D*) at any timepoint (t1) is defined as the difference between positions of the AIB neurite at that timepoint (t1) and the next timepoint (t2) (for which position was measured), divided by the time interval (t2-t1). These measurements are performed with CytoSHOW. To pseudocolor the neurites for representation, we used the same steps described in 'Confocal imaging of *C. elegans* larvae and image processing.'.

## Quantification of the angle of exit of the developing AIB distal neurite with the ventral turn of the nerve ring in the posterior neighborhood

The angle of exit (α) of the developing AIB distal neurite is measured as the angle between straight line tangents drawn along the separating distal segment of AIBL and the proximal neurite of AIBR and vice versa. These measurements are performed on deconvolved maximum intensity projections of diSPIM images where the neurons are oriented in an axial view. The angle of ventral turn of the nerve ring (β) is measured as the angle between straight line tangents drawn along segments of the nerve ring on either side of the ventral bend of the nerve ring in the posterior neighborhood (see *Figure 2— figure supplement 2I,J*). β is measured from images of embryos with proximal neighborhood labeled with *nphp-4* promoter (see Results and http://promoters.wormguides.org). All measurements are performed using CytoSHOW.

## Imaging and representation of synaptic protein RAB-3 in AIB in embryos

Time-lapse imaging of presynaptic proteins RAB-3, CLA-1, and SYD-2 in AIB in embryos was performed using diSPIM (*Wu et al., 2013*). To visualize the distribution of RAB-3 and CLA-1 along the neurite we straightened the distal neurite of each AIB neuron from maximum intensity projections where the AIB neurons are oriented in the axial (*Figure 7*) and lateral view (*Figure 7—figure supplement 1*), respectively.

## Quantification of RAB-3 distribution in *Syg-1(ky652)* larvae

The line intensity plot in *Figure 7—figure supplement 3M* was constructed in Fiji by drawing free-hand line selections along adhered and detached regions of the AIB neurite and using the Analyze > Plot Profile function. The mean intensities along 'Adhered' and 'Detached' regions were subtracted from the corresponding mean cell body intensities and plotted in *Figure 7—figure supplement 3N*.

## Quantification of nerve ring width from larval stage animals

The nerve ring was visualized using a 5.6 kb promoter of *cnd-1* (*Shah et al., 2017*) driving membrane-targeted GFP (PH:GFP) in wildtype and *syg-1(ky652)* mutant animals. Measurements were done on confocal image stacks where the worms are oriented on their side (z-stacks acquired along left-right axis of the worm, producing a lateral view of the neurons). On each side of the worm, a straight line selection along the anterior-posterior axis from one edge of the labeled nerve ring to the other was defined as the nerve ring width.

## Quantification of length of the dorsal midline shift (chiasm) from EM images

From a segmented EM dataset of the L4 larva JSH (*Brittin et al., 2018*; *White et al., 1986*), we calculated the number of z-slices containing segmented regions of the anterior-posterior shift (that forms the chiasm) of AIBL. We multiplied this number with the z-spacing of the dataset (60 nm) to obtain the anterior-posterior distance that the AIBL shift spans ($d_z$). We then calculated the x-y distance between the segmented regions of the AIBL shift in the topmost and bottommost z-slice($d_{x-y}$). We calculate the length of the shift in 3D (l) using the formula

$$l = \sqrt{d_z^2 + d_{x-y}^2}$$

The same measurements were repeated for the length of the dorsal midline shift of AIBR.

## Statistical analyses

Statistical analyses were conducted with PRISM seven software. For each case, the chosen statistical test is described in the figure legend and 'n' values are reported. Briefly, for continuous data, comparisons between two groups were determined by unpaired two-tailed t-test and comparisons within multiple groups were performed by ordinary one-way ANOVA. Error bars were reported as standard error of the mean (SEM). For categorical data, groups were compared with two-sided Fisher's exact test. The range of p-values for significant differences are reported in the figure legend. The Cohen's d statistic was determined for comparisons between continuous datasets with statistically significant differences, to obtain estimates of effect sizes.

## Acknowledgements

We thank Kang Shen, Harald Hutter and John Murray for providing strains and constructs. We thank Scott Emmons, Steve Cook, and Chris Brittin, and Mei Zhen and Daniel Witvliet for sharing their segmented EM data and adjacencies. We thank Thierry Emonet and members of the Colón-Ramos lab for help, advice and insightful comments during manuscript preparation. We thank Sarah Se-Hyun Jho and Kenya Collins for their contributions to the project. We thank the *Caenorhabditis* Genetics Center (funded by NIH Office of Research Infrastructure Programs P40 OD010440) for *C. elegans* strains. We thank the Research Center for Minority Institutions program, the Marine Biological Laboratories (MBL), and the Instituto de Neurobiología de la Universidad de Puerto Rico for providing meeting and brainstorming platforms. HS and DAC-R acknowledge the Whitman and Fellows program at MBL for providing funding and space for discussions valuable to this work. Research in the DAC-R, WAM, and ZB labs were supported by NIH grant No. R24-OD016474. MWM was supported by NIH by F32-NS098616. Research in HS lab was further supported by the intramural research program of the National Institute of Biomedical Imaging and Bioengineering (NIBIB), NIH. Research in ZB lab was further supported by an NIH center grant to MSKCC (P30CA008748). Research in the DAC-R lab was further supported by NIH R01NS076558, DP1NS111778 and by an HHMI Scholar Award. This research was also funded in part by the Gordon and Betty Moore foundation.

## Additional information

### Funding

| Funder | Grant reference number | Author |
|---|---|---|
| National Institutes of Health | R24-OD01647 | Zhirong Bao<br>William Mohler<br>Daniel A Colón-Ramos |
| National Institutes of Health | R01NS076558 | Daniel A Colón-Ramos |
| National Institutes of Health | DP1NS111778 | Daniel A Colón-Ramos |
| Howard Hughes Medical Institute | Faculty Scholar Award | Daniel A Colón-Ramos |
| Marine Biological Laboratory | Whitman and Fellows program | Hari Shroff<br>Daniel A Colón-Ramos |
| Gordon and Betty Moore Foundation | Moore Grant | Hari Shroff<br>Daniel A Colón-Ramos |
| Gruber Foundation | Gruber Science Fellowship | Titas Sengupta |
| National Institutes of Health | Predoctoral Training Program in Genetics NIH 2020 T32 GM. | Noelle L Koonce |
| National Institutes of Health | F32-NS098616 | Mark W Moyle |

| Funder | Grant reference number | Author |
|---|---|---|
| National Institutes of Health | NIBIB Intramural Research Program | Hari Shroff |
| National Institutes of Health | P30CA008748 | Zhirong Bao |

The funders had no role in study design, data collection and interpretation, or the decision to submit the work for publication.

## Author contributions

Titas Sengupta, Conceptualization, Data curation, Formal analysis, Investigation, Methodology, Resources, Software, Validation, Visualization, Writing - original draft, Writing - review and editing; Noelle L Koonce, Data curation, Methodology, Resources, Software, Visualization, Writing - review and editing; Nabor Vázquez-Martínez, Formal analysis, Methodology, Resources, Writing - review and editing; Mark W Moyle, Data curation, Methodology, Resources, Software, Writing - review and editing; Leighton H Duncan, Methodology, Resources, Software, Writing - review and editing; Sarah E Emerson, Formal analysis, Visualization, Writing - review and editing; Xiaofei Han, Investigation, Methodology, Software; Lin Shao, Li Fan, Resources; Yicong Wu, Formal analysis, Methodology, Software, Visualization, Writing - review and editing; Anthony Santella, Resources, Software; Zhirong Bao, Funding acquisition, Methodology, Supervision; William A Mohler, Data curation, Formal analysis, Funding acquisition, Methodology, Software, Supervision, Visualization; Hari Shroff, Funding acquisition, Methodology, Supervision, Visualization, Writing - review and editing; Daniel A Colón-Ramos, Conceptualization, Funding acquisition, Project administration, Supervision, Visualization, Writing - original draft, Writing - review and editing

## Author ORCIDs

Titas Sengupta ⓘ http://orcid.org/0000-0002-7228-719X
Zhirong Bao ⓘ http://orcid.org/0000-0002-2201-2745
Daniel A Colón-Ramos ⓘ http://orcid.org/0000-0003-0223-7717

## Decision letter and Author response

Decision letter https://doi.org/10.7554/eLife.71171.sa1
Author response https://doi.org/10.7554/eLife.71171.sa2

# Additional files

## Supplementary files

• Supplementary file 1. Plasmids generated and used in this study.
• Transparent reporting form

## Data availability

All data generated or analysed during this study are included in the manuscript and supporting files. Source data files have been provided for all plots in which individual data points are not represented - Figure 4N, Figure 6K, Figure 1—figure supplement 2I, Figure 1—figure supplement 2J, Figure 5—figure supplement 4N, Figure 6—figure supplement 1C, Figure 7—figure supplement 3N.

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

## Appendix 1

### Biophysical equations for retrograde zippering and unzippering

Force balance equations to derive velocities at the zippering and unzippering points

According to the biophysical models proposed for tissue culture cells (*Smít et al., 2017*), zippering and unzippering of neurite shafts occur as a result of primarily two forces acting on the neurite: adhesion, S and tension, T. Additionally as the neurite zippers or unzippers, it experiences frictional forces that act opposite to the direction of movement of the neurite, that is opposite to the velocity of the neurite.

At the zippering fork (see schematic above), adhesion acts in the direction of zippering, that is (in the direction of the zippering velocity $v_{zip}$) and favors zippering, while tension and friction act in the opposite direction, disfavoring zippering (see *Appendix 1—figure 1*). Assuming dynamic equilibrium (*Smít et al., 2017*), all the forces (with the+ or – signs representing their directions) add up to 0, resulting in the following equation:

$$S_{anterior} + T_{anterior}cos\theta - T_{anterior} - \eta v_{zip} = 0 \tag{1}$$

where $v_{zip}$ = velocity of retrograde zippering, $S_{anterior}$ = adhesion of the AIB neurite to the anterior neighborhood, $T_{anterior}$ = mechanical tension along the neurite, $\eta v_{zip}$ = friction forces at the zipper fork and $\theta$ = zipper angle.

Therefore, zippering velocity, $v_{zip}$ would be given by:

$$v_{zip} = \frac{S_{anterior}}{\eta} - \frac{T_{anterior}}{\eta}\left(1 - cos\theta\right) \tag{2}$$

On the other hand, at the unzippering fork (see schematic above), adhesion and friction act in the direction opposite to unzippering (i.e., opposite to the unzippering velocity $v_{unzip}$) and disfavors unzippering, while tension acts in the same direction favoring unzippering (see *Appendix 1—figure 1*). Again, assuming dynamic equilibrium (*Smít et al., 2017*), all the forces (with the+ or – signs representing their directions) add up to 0, resulting in the following equation: (*Smít et al., 2017*)

$$S_{posterior} + T_{posterior}cos\varphi - T_{posterior} + \eta v_{unzip} = 0 \tag{3}$$

Therefore, the velocity of unzippering from the posterior neighborhood, $v_{unzip}$, would be given by:

$$v_{unzip} = \frac{-S_{posterior}}{\eta} + \frac{T_{posterior}}{\eta}\left(1 - cos\varphi\right) \tag{4}$$

where $v_{unzip}$ = velocity of unzippering, $S_{posterior}$ = adhesion of the AIB neurite to the posterior neighborhood, $T_{posterior}$ = mechanical tension along the neurite, $\eta v_{unzip}$ = friction forces at the unzippering fork and $\varphi$ = zipper angle.

Since the nerve bundles constituting the posterior and anterior neighborhoods are parallel near the dorsal midline, therefore,

$$\theta = \varphi \tag{5}$$

Combining *equations (2), (4) and (5)*, we get,

$$v_{zip} + v_{unzip} = \frac{S_{anterior} - S_{posterior}}{\eta} - \frac{T_{anterior} - T_{posterior}}{\eta}(1 - cos\theta) \tag{6}$$

Since the same stretch of the AIB neurite that zippers onto the anterior neighborhood, concurrently unzippers from the posterior neighborhood (*Figure 3C*), and assuming mechanical tension is uniformly redistributed along the neurite (*Smít et al., 2017*), tension at the zippering and unzippering forks would be equal:

$$T_{anterior} = T_{posterior} \tag{7}$$

Combining (6) and (7) we obtain the following equation:

$$v_{zip} + v_{unzip} = \frac{(S_{anterior} - S_{posterior})}{\eta} \tag{8}$$

Since $v_{zip} > 0$ and $v_{unzip} > 0$, therefore,

$$S_{anterior} - S_{posterior} > 0, \vee S_{anterior} > S_{posterior}$$

The model, therefore, predicts the existence of differential adhesion which would result in forces driving adhesion to the anterior neighborhood.

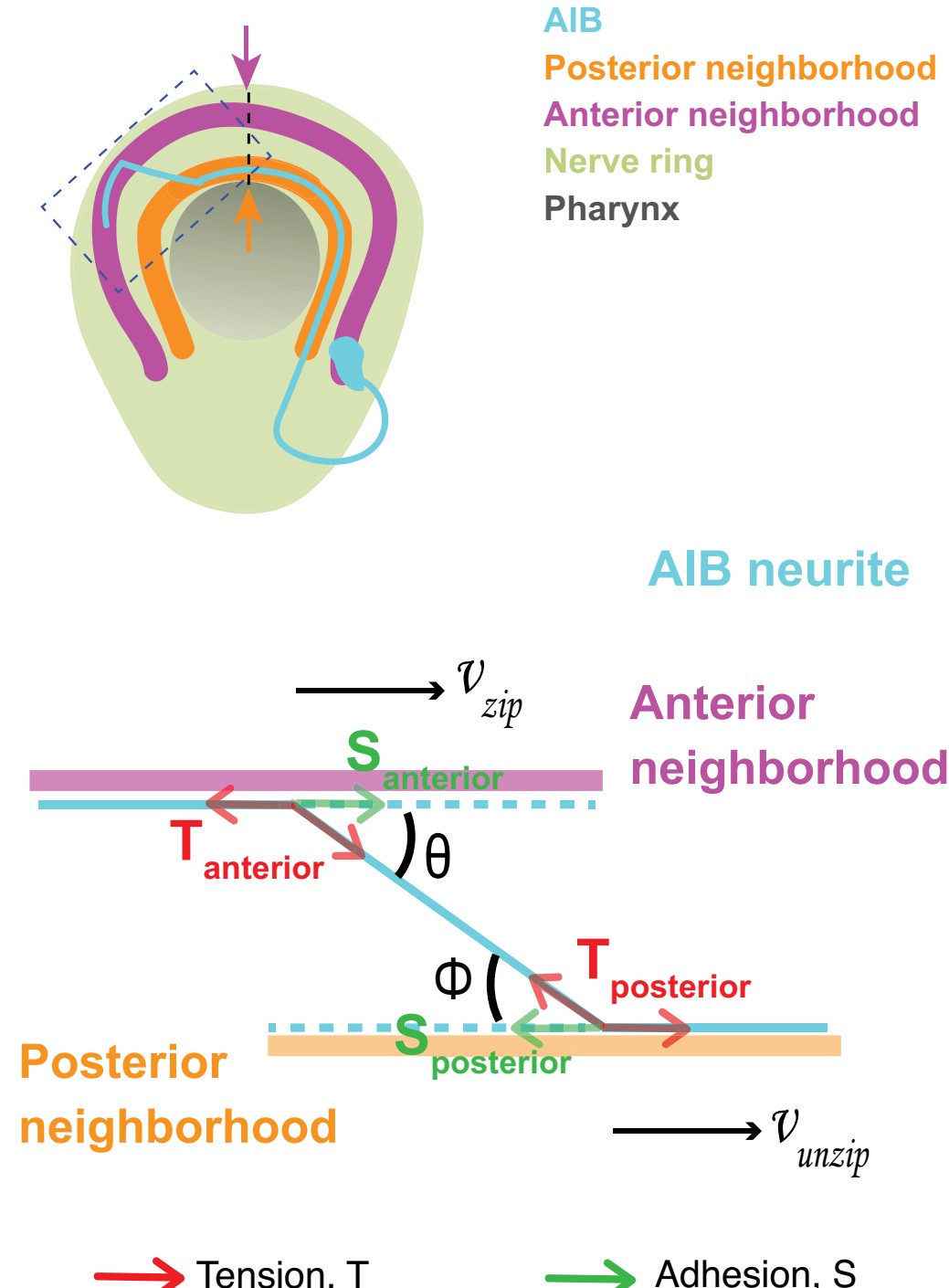

**Appendix 1—figure 1.** Biophysics of retrograde zippering and unzippering. Force balance at the zippering and unzippering points.

