## [Editor Report]

Your work provides novel and interesting insights into circuit formation, demonstrating how synaptic specificity is controlled at least in part by different cell adhesion during neurite placement. The revisions of your paper have addressed the points raised by the reviewers and we are glad to see that those revisions have further strengthened the conclusion of this paper.

---

## [Decision Letter]

**Decision letter after peer review:**

Thank you for submitting your article "Differential adhesion regulates neurite placement via a retrograde zippering mechanism" for consideration by *eLife*. Your article has been reviewed by 3 peer reviewers, one of whom is a member of our Board of Reviewing Editors, and the evaluation has been overseen by Piali Sengupta as the Senior Editor. The following individual involved in review of your submission has agreed to reveal their identity: Kang Shen (Reviewer #3).

Essential revisions:

After discussion among the three reviewers, the only experiment that two of the reviewers considered important to add is an analysis of an active zone marker (e.g. syd-2) to strengthen the conclusion that synapse formation only occurs after zippering. Currently, the way you discuss the zippering event with synapse formation is vague. If a marker like SYD-2 shows the same time course as RAB-3, then, a strong statement would be made that zippering happens before synapse formation and functions as a separate specificity mechanism. The reviewers think this is not a hard experiment to do and it will also validate the time course of GFP::RAB-3.

The other experimental request voiced by reviewer #1, regarding an improvement of the syg-1/2 expression analysis was, after discussion, deemed to be considered essential. Further discussion in the text that may assuage the concerns raised by reviewer #1 may be warranted.

All other requested revisions are editorial/textual/organizational in nature, but we ask you to nevertheless take them very seriously since several of them deal with important quantification issues.

*Reviewer #1:*

In their manuscript, Sengupta et al. describe a developmental mechanism that positions a single neuron across multiple layers in the hierarchical *C. elegans* nerve ring. The authors show that neighborhood placement of the interneuron AIB is established during embryogenesis and is maintained throughout development. AIB is one of the few *C. elegans* neurons that are divided into distinct pre- and post-synaptic regions, and its axons curiously occupy two physically separated neighborhoods or layers. How this occurs is not known. This study uses time-lapse imaging to show that unlike canonical axon tip outgrowth mediating fasciculation in a target region, AIB's axon occupies two neighborhoods by first growing completely into one, and then gradually unzippering from the first, switching, and zippering onto the second neighborhood. Importantly, axon outgrowth and neighborhood choice are continuously visualized during embryogenesis, an impressive experiment typically constrained by lack of cell-specific reporters during early development as well as the struggle of imaging embryos.

The authors posit that zippering is mediated by temporally regulated differential adhesive forces between AIB's neighboring pre- and post-synaptic neurons. How this differs from differential adhesion in classic fasciculating neurons is described but could be made much clearer. They proceed to identify the immunoglobulin syg-1/syg-2 receptor-ligand pair to be necessary and sufficient for AIB's axon switch; in syg-1/syg-2 mutants, AIB is not able to position itself in the second neighborhood and remains fasciculated with the first one, suggesting that adhesive forces are dampened in syg-1/syg-2 mutants. Lastly, the authors show that pre-synapse assembly follows zippering, linking AIB axon placement with synaptogenesis, and that this is also compromised in syg mutants.

The pipeline used to study axon outgrowth at a single-cell level in the embryo at relevant time points is commendable and will be useful to people studying *C. elegans* nervous system establishment. Although the overall manuscript and data are well-presented, we think the mechanism of retrograde zippering could be better described. Also, syg-1/syg-2 expression needs to be delineated to support the notion of differential adhesion between neighborhoods.

Major revisions:

1. The process of zippering can be described in more detail and contrasted with canonical axon tip outgrowth and selective fasciculation, especially in the context of differential adhesion and biophysical forces mediating cell adhesion. Drawing on more literature that describes zippering in other contexts might be helpful.

2. Figure 5: syg-1 expression changes and enrichment in the anterior neighborhood warrant better characterization. The authors write that "syg-1 transcriptional reporter shows banded expression in ~20 neurons present in the AIB posterior and anterior neighborhoods, with specific enrichment in anterior neighborhood" but identities of these neurons are not characterized using the transcriptional reporter. This is relevant because if there are more neurons in one neighborhood compared to the other, then you would naturally see more expression (given the method of quantification) but it wouldn't necessarily be "enrichment" since more neurons are growing into that neighborhood at that point in development and those additional neurons might not be synapsing with AIB. Or they might be the only neurons synapsing with and thus attracting AIB, in which case differential levels of syg-1 are not as important as presence of it. Understandably, quantifying embryonic expression at the single-cell level is not trivial in the absence of cell specific reporters in the background, but could the authors characterize expression at L1 (since the enrichment and AIB axon placement are maintained)? Could the authors use an endogenous nuclear-localized reporter for syg-1 to aid quantification and glean cell identities, given candidates from Packer et al. data and their own experiments?

3. Related to point 1, enrichment of translational SYG-1 reporter in the anterior neighborhood is not obvious.

4. [line 365] The authors state that "SYG-1 expression in RIM neurons contributes to AIB neurite placement" but Packer et al. data oddly does not show a temporally dynamic enrichment in RIM (supp table 1) at the time AIB switches neighborhoods. Could it be that AIB only switches to the anterior neighborhood once the target neuron RIM grows into it? If so, it's unclear why this would be termed enrichment. The authors need a more robust readout for temporal syg-1 expression in RIM.

*Reviewer #2:*

A large amount of data is presented in this paper. The experiments are carefully documented and support the conclusions. Of particular importance is the live imaging of the outgrowth of the AIB neurite in the embryo. This is challenging and required the development of a new marker for labelling and the adaptation of a new type of microscope. This enabled the initial and surprising observation that part of the neurite relocates after outgrowth. I'm not sure that the mathematical modeling adds much. The main conclusion is that the modeling is consistent with a "net increase of adhesive forces in the anterior neighbourhood", which is to be expected. The authors then try to identify the relevant adhesion molecules and find that a pair of IgCAMs (syg-1 and syg-2), which are known to act as receptor-ligand pair, are involved. A series of experiments establishes that syg-2 act in the AIB neurons, whereas syg-1 does not. The neurite positioning defects in syg-1 and syg-2 mutants are partially penetrant, suggesting that other adhesion molecules must be involved. While a large percentage of mutant animals show defects, the defects within an individual animal are surprisingly low with only 21.5% +/- 4% of the neurite detached. This would suggest that syg-1/syg-2 aren't even the major adhesion molecules involved here. In further studies, where the authors ablate the RIM neurons (which express syg-1), the authors use a different measure to quantify the defects (minimal distance between neurite segments, Suppl Figure 7). This makes it difficult to compare the results to those of the syg-1 mutants. For the ectopic expression experiments with syg-1 the authors only report the percentage of animal with defects and not the extent of the defects (how much of the neurite was in an abnormal position).

Overall, this is a very detailed study describing an important novel mechanism for neurite positioning within an nerve bundle.

*Reviewer #3:*

This is a very interesting manuscript describing the changes of neurite position in a complex neuropil during development. The experimental system is well chosen because AIB's function within the circuit requires its neurite to be in two different neuropil "neighborhoods". The manuscript included some technically difficult experiments of imaging neurite outgrowth in *C. elegans* embryos which are very hard to do. The surprising finding here is that neurite position is not sole dependent on its growth cone navigation. In the case of the AIB neuron, the growth cone is anchored after it reaches its destination point and then a segment of the neurite shift direction towards its final position through a zippering action. They also show that this shift in position is driven by adhesion molecules SYG-1 and SYG-2. Overall, I think this is a strong candidate for *eLife*.

My main point is about the relationship between synapse formation and neurite zippering. In my opinion, this is an interesting point because it would tell us if the zippering behavior is a consequence of synapse formation or it is a distinct specificity step before synapse formation. From the time course that was described in the paper, it seems that the accumulation of RAB-3 only starts after the zippering has completed. I would suggest the authors to examine at least another synaptic marker like SNB-1 or SYD-2. We have created cell specific endogenous labeling of several active zone markers that can be used for these experiments. If the results hold, then, I think the authors should make it clear in the text that the zippering takes place before synapse formation and serves as a distinct step in achieving the neighborhood specificity.

Examine the localization of at least one active zone proteins and compare the time course of synapse formation to the zippering of AIB neurite.

---

## [Author Response]

Reviewer #1:In their manuscript, Sengupta et al. describe a developmental mechanism that positions a single neuron across multiple layers in the hierarchical *C. elegans* nerve ring. The authors show that neighborhood placement of the interneuron AIB is established during embryogenesis and is maintained throughout development. AIB is one of the few C. elegans neurons that are divided into distinct pre- and post-synaptic regions, and its axons curiously occupy two physically separated neighborhoods or layers. How this occurs is not known. This study uses time-lapse imaging to show that unlike canonical axon tip outgrowth mediating fasciculation in a target region, AIB's axon occupies two neighborhoods by first growing completely into one, and then gradually unzippering from the first, switching, and zippering onto the second neighborhood. Importantly, axon outgrowth and neighborhood choice are continuously visualized during embryogenesis, an impressive experiment typically constrained by lack of cell-specific reporters during early development as well as the struggle of imaging embryos.The authors posit that zippering is mediated by temporally regulated differential adhesive forces between AIB's neighboring pre- and post-synaptic neurons. How this differs from differential adhesion in classic fasciculating neurons is described but could be made much clearer. They proceed to identify the immunoglobulin syg-1/syg-2 receptor-ligand pair to be necessary and sufficient for AIB's axon switch; in syg-1/syg-2 mutants, AIB is not able to position itself in the second neighborhood and remains fasciculated with the first one, suggesting that adhesive forces are dampened in syg-1/syg-2 mutants. Lastly, the authors show that pre-synapse assembly follows zippering, linking AIB axon placement with synaptogenesis, and that this is also compromised in syg mutants.The pipeline used to study axon outgrowth at a single-cell level in the embryo at relevant time points is commendable and will be useful to people studying *C. elegans* nervous system establishment. Although the overall manuscript and data are well-presented, we think the mechanism of retrograde zippering could be better described. Also, syg-1/syg-2 expression needs to be delineated to support the notion of differential adhesion between neighborhoods.

We have further clarified the novelty of the zippering mechanism, contrasting it with tip-directed outgrowth. We have also performed a thorough analysis of *syg-1* expression (detailed below, new Figure 5—figure supplement 2).

Major revisions:1. The process of zippering can be described in more detail and contrasted with canonical axon tip outgrowth and selective fasciculation, especially in the context of differential adhesion and biophysical forces mediating cell adhesion. Drawing on more literature that describes zippering in other contexts might be helpful.

We have added new textual clarification on how differential adhesion gives rise to the observed zippering mechanism, and distinguished this mechanism from canonical tip-directed outgrowth (, all line numbers refer to the lines in the annotated Word document files). While retrograde zippering had been described for tissue culture cells, it had not been documented in vivo, and its importance, if any, remained unknown. We now better explain in the text that our study is, to our knowledge, the first in vivo demonstration of the role of retrograde zippering in neurite placement during development.

2. Figure 5: syg-1 expression changes and enrichment in the anterior neighborhood warrant better characterization. The authors write that "syg-1 transcriptional reporter shows banded expression in ~20 neurons present in the AIB posterior and anterior neighborhoods, with specific enrichment in anterior neighborhood" but identities of these neurons are not characterized using the transcriptional reporter. This is relevant because if there are more neurons in one neighborhood compared to the other, then you would naturally see more expression (given the method of quantification) but it wouldn't necessarily be "enrichment" since more neurons are growing into that neighborhood at that point in development and those additional neurons might not be synapsing with AIB. Or they might be the only neurons synapsing with and thus attracting AIB, in which case differential levels of syg-1 are not as important as presence of it. Understandably, quantifying embryonic expression at the single-cell level is not trivial in the absence of cell specific reporters in the background, but could the authors characterize expression at L1 (since the enrichment and AIB axon placement are maintained)? Could the authors use an endogenous nuclear-localized reporter for syg-1 to aid quantification and glean cell identities, given candidates from Packer et al. data and their own experiments?

We identify the SYG-1 expressing neurons in embryos by performing single-cell lineage analysis as described (Bao et al., 2006) and identify sixteen SYG-1 expressing neurons in the nerve ring: six in the anterior neighborhood, and ten in the posterior neighborhood (Figure 5—figure supplement 2). Our findings, which are consistent with transcriptomic datasets (Packer et al., 2019), extends our understanding of SYG-1 nerve ring expression during development, and are consistent with our observations on dynamic *syg-1* expression levels that regulate AIB neurite transition between neighborhoods by zippering. We present these findings in new Figure 5—figure supplement 2 and discuss them in lines 354-361 and 1536-1565.

3. Related to point 1, enrichment of translational SYG-1 reporter in the anterior neighborhood is not obvious.

We have revised Figure 5, including a timepoint before zippering (Figure 5L) and one during zippering (Figure 5M), to more clearly represent the increase of SYG-1 in the anterior neighborhood.

4. [line 365] The authors state that "SYG-1 expression in RIM neurons contributes to AIB neurite placement" but Packer et al. data oddly does not show a temporally dynamic enrichment in RIM (supp table 1) at the time AIB switches neighborhoods. Could it be that AIB only switches to the anterior neighborhood once the target neuron RIM grows into it? If so, it's unclear why this would be termed enrichment. The authors need a more robust readout for temporal syg-1 expression in RIM.

We use the term “SYG-1 enrichment” to refer to increase in SYG-1 expression levels in the entire anterior neighborhood. These increases can be caused by (i) ingrowth of SYG-1-expressing neurons into the neighborhood and (ii) increase in SYG-1 levels in neurons already in the neighborhood. The RIM neurons grow into the anterior neighborhood, contributing to the local increase of overall SYG-1 levels in this neighborhood. We have clarified this in the text (lines 377-378) and also now provide a more extensive characterization of the SYG-1 expressing neurons in embryos (as described in Comment 2 above).

Reviewer #2:A large amount of data is presented in this paper. The experiments are carefully documented and support the conclusions. Of particular importance is the live imaging of the outgrowth of the AIB neurite in the embryo. This is challenging and required the development of a new marker for labelling and the adaptation of a new type of microscope. This enabled the initial and surprising observation that part of the neurite relocates after outgrowth. I'm not sure that the mathematical modeling adds much. The main conclusion is that the modeling is consistent with a "net increase of adhesive forces in the anterior neighbourhood", which is to be expected. The authors then try to identify the relevant adhesion molecules and find that a pair of IgCAMs (syg-1 and syg-2), which are known to act as receptor-ligand pair, are involved. A series of experiments establishes that syg-2 act in the AIB neurons, whereas syg-1 does not. The neurite positioning defects in syg-1 and syg-2 mutants are partially penetrant, suggesting that other adhesion molecules must be involved. While a large percentage of mutant animals show defects, the defects within an individual animal are surprisingly low with only 21.5% +/- 4% of the neurite detached. This would suggest that syg-1/syg-2 aren't even the major adhesion molecules involved here. In further studies, where the authors ablate the RIM neurons (which express syg-1), the authors use a different measure to quantify the defects (minimal distance between neurite segments, Suppl Figure 7). This makes it difficult to compare the results to those of the syg-1 mutants. For the ectopic expression experiments with syg-1 the authors only report the percentage of animal with defects and not the extent of the defects (how much of the neurite was in an abnormal position).Overall, this is a very detailed study describing an important novel mechanism for neurite positioning within an nerve bundle.

We have added in this revised version additional quantifications, including ‘the minimal distance between neurite segments’ measure (the one used for the RIM ablation experiments) in Figure 4—figure supplement 1 for the placement defects in *syg-1(ky652)* and *syg-2(ky671)* mutants, allowing direct comparisons between the phenotypes. We have also added a measure for the percentage of the distal neurite that is mispositioned in the ectopic expression experiments (Figure 6—figure supplement 1A). We do not claim that SYG-1/SYG-2 are the only adhesion molecules involved in AIB neurite placement, but that they are required for complete and proper placement of the neurite. We clarify this in the text.

Reviewer #3:This is a very interesting manuscript describing the changes of neurite position in a complex neuropil during development. The experimental system is well chosen because AIB's function within the circuit requires its neurite to be in two different neuropil "neighborhoods". The manuscript included some technically difficult experiments of imaging neurite outgrowth in *C. elegans* embryos which are very hard to do. The surprising finding here is that neurite position is not sole dependent on its growth cone navigation. In the case of the AIB neuron, the growth cone is anchored after it reaches its destination point and then a segment of the neurite shift direction towards its final position through a zippering action. They also show that this shift in position is driven by adhesion molecules SYG-1 and SYG-2. Overall, I think this is a strong candidate for eLife.My main point is about the relationship between synapse formation and neurite zippering. In my opinion, this is an interesting point because it would tell us if the zippering behavior is a consequence of synapse formation or it is a distinct specificity step before synapse formation. From the time course that was described in the paper, it seems that the accumulation of RAB-3 only starts after the zippering has completed. I would suggest the authors to examine at least another synaptic marker like SNB-1 or SYD-2. We have created cell specific endogenous labeling of several active zone markers that can be used for these experiments. If the results hold, then, I think the authors should make it clear in the text that the zippering takes place before synapse formation and serves as a distinct step in achieving the neighborhood specificity.

We thank the reviewer for generously sending us strains for cell-specific endogenous active zone protein labeling (McDonald et al., 2021) using the SapTrap method (Schwartz and Jorgensen, 2016). We made constructs expressing FLP recombinase downstream of the *inx-1* and *unc-42* promoters for cell-specific labeling of these active zone proteins in AIB and injected them into the strains. Although we observed cell-specific synaptic signal in larvae with both *inx-1p* and *unc-42p*-driven FLP, we were not able to observe signal during embryogenesis, probably due to cell-specific synaptic protein expression levels being low.

Therefore and to address the reviewer’s question about the temporal order of zippering and synapse formation, we have cell-specifically expressed two active zone proteins in AIB (CLA-1 and SYD-2) and measured their intensities over time in AIB in embryos (Figure 7, Figure 7—figure supplement 1A-F). We find that similar to RAB-3, synaptic signal is not visible until after the end of zippering, and progressively increases over time following zippering. These observations suggest that synapses do not initiate retrograde zippering. We added the time-course of active zone protein localization in AIB in the context of the time course of retrograde zippering (Figure 7J). Consistent with these observations, in a *syd-2(ola341)* allele identified in our screen, we find that although synapses are mislocalized, AIB neurite placement is unaffected, consistent with the idea that synapse formation is not upstream of zippering-mediated placement (Figure 7—figure supplement 1G-K).

We acknowledge, however, that our studies are limited by detection of the synaptic proteins, and that, while it does not appear that synaptogenesis leads to zippering, a cooperative and synergistic relationship might exist between the process of zippering and synaptogenesis to hold the neurite position in place. We have added text to better discuss this relationship between zippering and synaptogenesis in light of these findings.

Examine the localization of at least one active zone proteins and compare the time course of synapse formation to the zippering of AIB neurite.

We have included the time course of localization for SYD-2 and CLA-1 and shown that they occur concurrently with RAB-3 localization (Figure 7, Figure 7—figure supplement 1) and after retrograde zippering of the AIB neurite (Figure 7, Figure 7—figure supplement 1), supporting our conclusions. We now better explain the model of how we envision the interaction between synaptogenesis and neurite placement in lines 421-440, 545-559.

References

Bao, Z., Murray, J. I., Boyle, T., Ooi, S. L., Sandel, M. J. and Waterston, R. H. 2006. Automated cell lineage tracing in *Caenorhabditis elegans*. Proceedings of the National Academy of Sciences of the United States of America.

Mcdonald, N. A., Fetter, R. D. and Shen, K. 2021. Author Correction: Assembly of synaptic active zones requires phase separation of scaffold molecules. *Nature,* 595**,** E35.

Packer, J. S., Zhu, Q., Huynh, C., Sivaramakrishnan, P., Preston, E., Dueck, H., Stefanik, D., Tan, K., Trapnell, C., Kim, J., Waterston, R. H. and Murray, J. I. 2019. A lineage-resolved molecular atlas of *C. elegans* embryogenesis at single-cell resolution. *Science,* 365.

Schwartz, M. L. and Jorgensen, E. M. 2016. SapTrap, a Toolkit for High-Throughput CRISPR/Cas9 Gene Modification in *Caenorhabditis elegans*. *Genetics,* 202**,** 1277-88.

White, J. G., Southgate, E., Thomson, J. N. and Brenner, S. 1986. The structure of the nervous system of the nematode *Caenorhabditis elegans*. *Philos Trans R Soc Lond B Biol Sci,* 314**,** 1-340.